# Wobble tRNA modification and hydrophilic amino acid patterns dictate protein fate

Francesca Rapino[1,2,3,4,14 ✉], Zhaoli Zhou[1,2,3,5,14], Ana Maria Roncero Sanchez[1,2,3,14], Marc Joiret[1,2,3,6], Christian Seca[1,2,3], Najla El Hachem [1,2,3], Gianluca Valenti[3,7], Sara Latini [1,2,3], Kateryna Shostak[2,3,8], Liesbet Geris [2,3,6], Ping Li[5], Gang Huang[5], Gabriel Mazzucchelli[2,3,9], Dominique Baiwir [2,3,9], Christophe J. Desmet[2,3,10,11], Alain Chariot[2,3,8,12], Michel Georges[2,3,11,13] & Pierre Close [1,2,3,12 ✉]

Regulation of mRNA translation elongation impacts nascent protein synthesis and integrity and plays a critical role in disease establishment. Here, we investigate features linking regulation of codon-dependent translation elongation to protein expression and homeostasis. Using knockdown models of enzymes that catalyze the $mcm^5s^2$ wobble uridine tRNA modification ($U_{34}$-enzymes), we show that gene codon content is necessary but not sufficient to predict protein fate. While translation defects upon perturbation of $U_{34}$-enzymes are strictly dependent on codon content, the consequences on protein output are determined by other features. Specific hydrophilic motifs cause protein aggregation and degradation upon codon-dependent translation elongation defects. Accordingly, the combination of codon content and the presence of hydrophilic motifs define the proteome whose maintenance relies on $U_{34}$-tRNA modification. Together, these results uncover the mechanism linking wobble tRNA modification to mRNA translation and aggregation to maintain proteome homeostasis.

[1] Laboratory of Cancer Signaling, University of Liège, Liège, Belgium. [2] GIGA-Institute, University of Liège, Liège, Belgium. [3] University of Liège, Liège, Belgium. [4] Department of Radiation Oncology (Maastro), GROW School for Oncology and Developmental Biology, Maastricht University Medical Centre+, Maastricht, The Netherlands. [5] Shanghai Key Laboratory of Molecular Imaging, Shanghai University of Medicine and Health Sciences, Shanghai, China. [6] Biomechanics Research Unit, University of Liège, Liège, Belgium. [7] Unité de Recherche Transitions, University of Liège, Liège, Belgium. [8] Laboratory of Medical Chemistry, University of Liège, Liège, Belgium. [9] Mass Spectrometry Laboratory, System Biology and Chemical Biology, University of Liège, Liège, Belgium. [10] Laboratory of Cellular and Molecular Immunology, University of Liège, Liège, Belgium. [11] Faculty of Veterinary Medicine, University of Liege, Liege, Belgium. [12] WELBIO, University of Liege, Liege, Belgium. [13] Unit of Animal Genomics, University of Liège, Liège, Belgium. [14]These authors contributed equally: Francesca Rapino, Zhaoli Zhou, Ana Maria Roncero Sanchez. ✉email: Francesca.Rapino@uliege.be; Pierre.Close@uliege.be

Adequate regulation of mRNA translation rate and speed is essential for cellular homeostasis. mRNA translation impacts protein output, but it also regulates co-translational processes that assist nascent polypeptides. The speed of ribosomes on mRNA during translation is not uniform and an increasing amount of evidence has demonstrated that translation kinetics play a prominent role in proteostasis by impacting co-translational protein folding, translation fidelity, or protein quality control pathways[1–9]. In addition, cells must not only promote accurate folding but also have to prevent accumulation of misfolded proteins that may arise from errors in translation, aberrant mRNAs, or defects in the chaperone machinery[10,11].

Local translation elongation rate is tightly regulated at the level of individual codons through the integration of different layers, including tRNA abundance and modification, tRNA selection, and codon usage. In this context, wobble uridine ($U_{34}$) tRNA modifications, including the addition of both methoxycarbonylmethyl ($mcm^5$) and thiol ($s^2$), are crucial for the decoding of specific codons, mainly AA-ending codons (i.e., AAA, GAA, CAA)[12–19]. Consequently, loss of these modifications was shown to lower expression of functionally important proteins whose mRNA is enriched in such codons, a process that can be rescued by the replacement of AA-ending codons by synonymous, $U_{34}$-tRNA modification independent, AG-ending codons[19,20]. The enzymatic pathway leading to the addition of the $mcm^5s^2$ modification at $U_{34}$-tRNAs ($U_{34}$-enzymes) encloses the acetyltransferase complex Elongator, a six subunit complex (ELP1-6) where ELP3 harbors the catalytic activity, the methyltransferase ALKBH8 (Alkylation repair homolog 8) and the thiouridylase complex composed of CTU1/2 proteins (Cytosolic thiouridylase proteins 1/2)[12,14,15,21,22]. Ribosome profiling experiments demonstrate that loss of $U_{34}$-enzymes leads to ribosome accumulation (i.e., translation elongation defects) mainly at AA-ending codons (i.e., AAA, GAA, CAA)[16–19]. However, the exact consequences of this codon-specific translation pausing on subsequent protein expression and homeostasis still remain elusive. Recent studies showed that loss of $U_{34}$-enzymes leads to increase protein aggregation[17,19]. However, the mechanisms linking codon-dependent translation dysregulation to protein aggregation remain poorly understood.

tRNA modifications emerge as key players in development and cancer. In particular, loss of the $U_{34}$-enzyme *Elp3* in murin cortical progenitors results in neurogenesis defects and microcephaly through induction of an unfolded protein response (UPR)[18]. Also, altered proteostasis consequent to *Elp3* loss affects the development of the cochlea and results in deafness[23]. Mutations in Elongator genes are associated with neurodegenerative disorders[24]. In cancer, we and others have recently uncovered a key role of $U_{34}$-enzymes in tumor development, metastasis and resistance to therapy by promoting efficient codon-specific translation. ELP3 was shown to drive Wnt-dependent intestinal tumor initiation by maintaining a subpopulation of Lgr5$^+$/Dclk1$^+$/Sox9$^+$ cancer stem cells[25]. Also, $U_{34}$-enzymes regulate breast cancer metastasis through the translation of the pro-invasive DEK oncoprotein[20]. Furthermore, $U_{34}$-enzymes promote resistance to targeted therapy in melanoma, through regulation of HIF1A mRNA translation in a codon-dependent manner[19]. Finally, Elongator and other $U_{34}$-enzymes were identified as a key determinant of gemcitabine sensitivity in gallbladder cancer by regulating the mRNA translation of hnRNPQ[26]. Together, these evidences highlight the role of $U_{34}$-enzymes in cancer by promoting the mRNA translation of key oncoproteins[22]. However, the extent of the proteome subset whose expression relies on the $U_{34}$-tRNA modification pathway remains to be determined.

Here, we took advantage of loss-of-function models of $U_{34}$-enzymes to investigate the impact of codon-dependent regulation of translation elongation on proteome expression and maintenance. Surprisingly, we find that gene codon content is necessary but not sufficient to predict protein fate upon defective $U_{34}$-tRNA modification. Patterns of hydrophilic pentasequences link defects in codon-dependent translation elongation to protein aggregation and subsequent degradation.

## Results

### Analysis of $U_{34}$-codons frequency across the human orfeome uncovers translational targets of the $U_{34}$-tRNA modification pathway.
Biological phenotypes associated with modulation of the $mcm^5s^2$ wobble uridine tRNA modification have been associated with decoding of specific codon during translation elongation, mainly of three codons AAA, GAA, and CAA ($U_{34}$-codons)[16–19]. We designed dual-luciferase constructs encoding six in frame repetitions of $U_{34}$-codons (xAA) upstream of NanoLuc and six in frame repetitions of the corresponding synonymous codon (xAG), whose decoding does not require $U_{34}$-tRNA modification, upstream of FireflyLuc (Fig. 1a and Supplementary Fig. 1a). Depletion of the $U_{34}$-enzymes, ELP3, or CTU2, strongly decreased NanoLuc activity but did not impact FireflyLuc activity (Fig. 1b and Supplementary Fig. 1b-d), and no effect was seen using a construct lacking the codon repetitions (Supplementary Fig. 1e-f). Together with previous evidences[19,20,27–29], these data indicate that decoding the -AA ending codons requires $U_{34}$-enzymes.

In order to identify the mechanisms linking codon-dependent translational optimization and protein fate, we computed the frequency bias across the human genome of the three $U_{34}$-codons (i.e., AAA, GAA, and CAA). To this end, we calculated a $q$ value for each $U_{34}$-codon in every gene of the human orfeome in order to define codon enrichment in a statistical manner ($q < 0.05$, see the "Methods" section). Interestingly, 4331 genes display a high frequency of at least one of the three codons ($q$ value <0.05; Supplementary Fig. 1g), among which 608 genes are strongly enriched in all three codons (Fig. 1c and Supplementary Fig. 1h). In line with previously published evidence in *S. cerevisae*, genes enriched in these codons significantly cluster in biological pathways related to the cell cycle[30,31] (Supplementary Fig. 1i, Supplementary Data1) as well as in specific protein families, including kinesins (Supplementary Fig. 1j). As expected, depletion of ELP3 and CTU2, two of the $U_{34}$-enzymes[12,13,22], leads to a strong increase in ribosome occupancy (i.e., defect in translation elongation) at the top five transcripts enriched in $U_{34}$-codons (i.e., CEP83, CEP290, HMMR, NEXN, and MNS1; Fig. 1d and Supplementary Fig. 1k). This correlates with a strong decrease in their protein expression levels (Fig. 1e). No difference in the expression levels of the corresponding mRNAs was detected (Supplementary Fig. 1l). These data indicate that high abundance of $U_{34}$-codons in transcripts predicts dependency toward $U_{34}$-enzymes for correct mRNA translation and corresponding protein expression.

### mRNA codon content is not sufficient to predict protein abundance upon loss of the $U_{34}$-enzymes.
In order to assess if the enrichment of $U_{34}$-codons in mRNAs is sufficient to predict their protein expression levels upon loss of $U_{34}$-enzymes, we performed RNA-seq and label-free quantitative proteomics upon ELP3 depletion (Fig. 2a and Supplementary Fig. 2a-b). Surprisingly, even though downregulated proteins were significantly enriched in $U_{34}$-codons in their corresponding mRNA as compared to the genome ORFs content ($\chi^2$ test, $p = 3.79 \times 10^{-19}$; Fig. 2a and Supplementary Data 2), a large number of genes enriched in $U_{34}$-codons remained unchanged in their protein expression (Fig. 2b and Supplementary Fig. 2c). Of note, no major

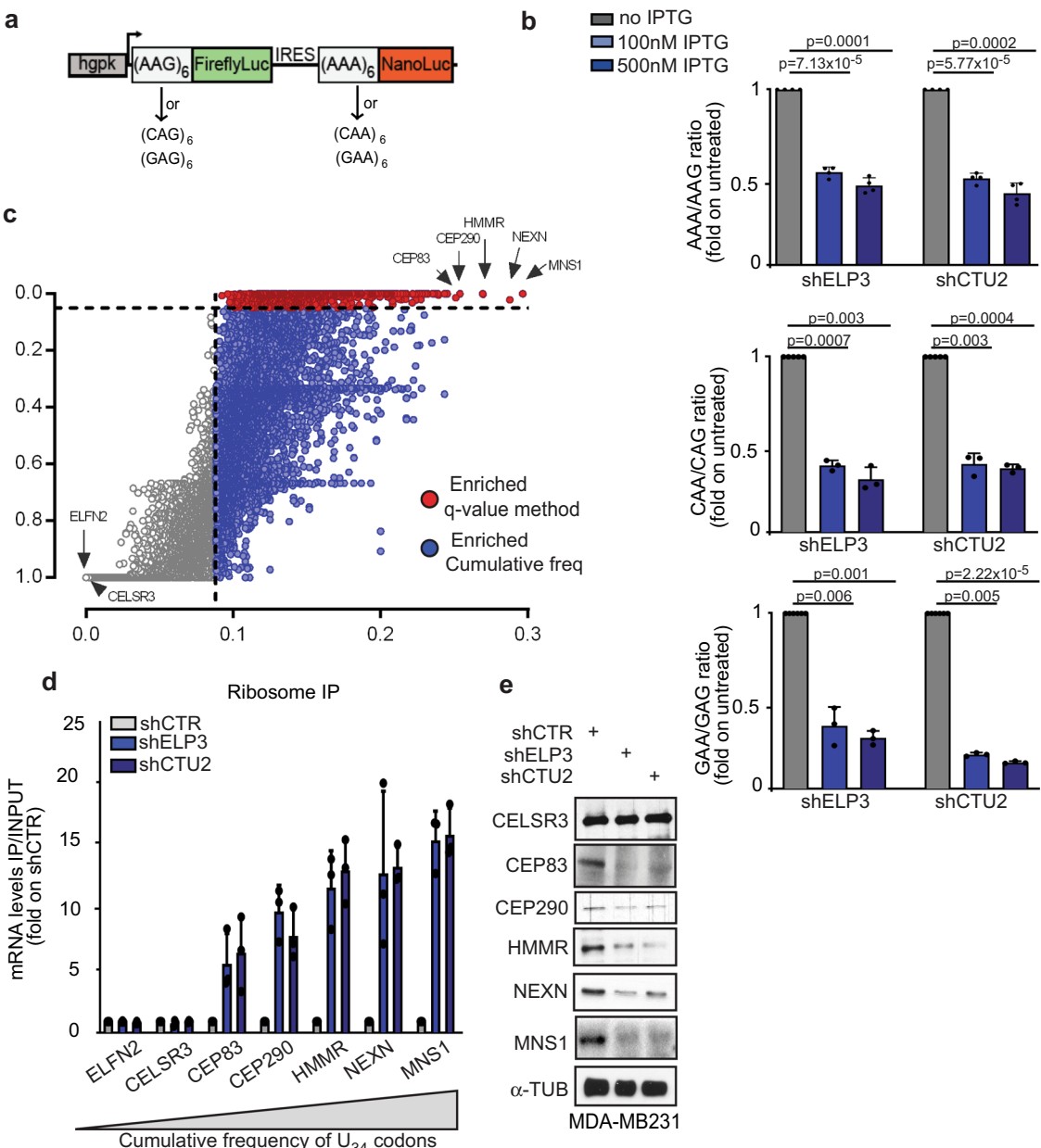

**Fig. 1 mRNA codon content predicts translation dependency on U$_{34}$-enzymes. a** Schematic representation of the dual-luciferase construct. **b** NanoLuc/ Firefly luminescence ratio of MDA-MB231 cells stably expressing the indicated constructs and depleted or not of ELP3 or CTU2 upon IPTG treatment. Data are normalized on non-treated conditions (IPTG 0 μM, $n = 3$–4 independent experiments, two-sided $t$-test, data are mean + s.d). **c** AAA, CAA, and GAA codon frequency and $q$ value were computed for each human gene, red and blue dots represent genes enriched in the three U$_{34}$-codons (AAA&GAA&CAA $q$ value <0.05 = red dots; AAA&GAA&CAA cumulative frequency >0.088 = blue dots). **d** qRT-PCR after ribosome immunoprecipitation in MDA-MDMB231 cells. $n = 2$ independent experiments, two-sided $t$-test, data are mean + s.d. **e** Levels of indicated proteins were determined in cells by western blot ($n = 1$ replicate).

changes at the mRNA steady-state levels were detected for U$_{34}$-enriched genes upon ELP3 depletion (Supplementary Fig. 2b and Supplementary Data 2). In agreement with our codon-content analysis, many members of the kinesin family were found downregulated at the protein, but not the mRNA, level in the absence of ELP3 (i.e., KIF4A, KIF14, KIF15, KIF13B, KIF5A, KIF5C, and KIF1B; gene family enrichment analysis, $p = 1.10 \times 10^{-13}$) (Fig. 2c and Supplementary Fig. 2d-e). Surprisingly, other kinesins, such as KIF5B or KIF23, even though strongly enriched in U$_{34}$-codons (i.e., KIF5B: 17.11% and KIF23: 13.4%), were not downregulated in ELP3-depleted cells (Fig. 2a, c and Supplementary Data 2). Strikingly, ribosomes invariably accumulate at

kinesin transcripts enriched in U$_{34}$-codons (i.e., KIF4A, KIF14, KIF15, KIF5B, and KIF23), independently of their protein expression levels, and with no change in the corresponding mRNA levels (Fig. 2d and Supplementary Fig. 2f). Therefore, the U$_{34}$-codon content of mRNAs is predictive of changes in ribosome translational elongation, but is not sufficient to predict protein expression output in the absence of U$_{34}$-enzymes.

**Loss of U$_{34}$-enzymes leads to protein aggregation in a codon-dependent manner.** Defective translation elongation has been associated with increased protein misfolding and aggregation[32]. Accordingly, loss of U$_{34}$-enzymes leads to increased protein

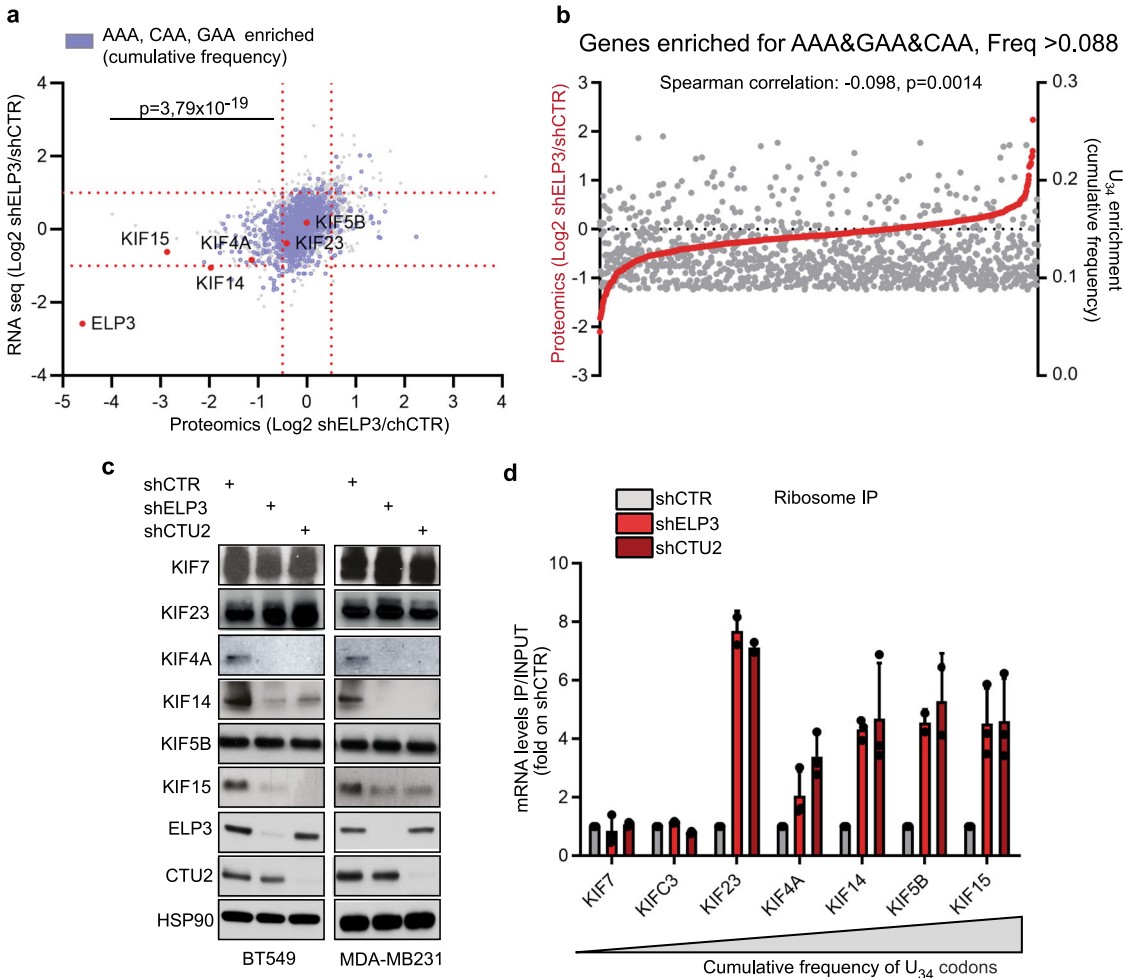

**Fig. 2 mRNA codon content poorly predicts protein expression fate upon loss of U$_{34}$-enzymes. a** Dot plot of RNA-seq and proteomics of BT549 cells depleted or not of ELP3 ($n = 3$ independent experiments); genes enriched in U$_{34}$-codons are shown in blue. U$_{34}$-codons enrichment in downregulated proteins was calculated by $\chi^2$ test (two-sided). **b**, Frequency of U$_{34}$-codons and protein expression in BT549 cells depleted of ELP3 was plotted, correlation was calculated by Spearman test (two-sided). **c** Levels of indicated proteins were determined in cells depleted of ELP3 or CTU2 by western blot ($n = 4$ replicates). **d** qRT-PCR after ribosome immunoprecipitation in MDA-MDMB231 cells. $n = 2$ independent experiments, data are mean + s.d.

aggregation in a variety of cell lines (Fig. 3a–c)[19,33]. We surmised that defects in ribosome kinetics upon loss of U$_{34}$-enzymes might result in defective protein expression only for proteins that are integrated into aggregates for clearance. To test this hypothesis, we purified the aggregates formed in ELP3-depleted cells and we assessed their protein content by quantitative proteomics (Fig. 3d). Strikingly, proteins exclusively detected in aggregates upon ELP3 depletion were strongly biased toward U$_{34}$-codons in their corresponding mRNAs ($p = 0.00024$ vs $p = 0.355$ in control aggregates compared to human orfeome). Also, previously identified targets of U$_{34}$-enzymes, KIF4A and KIF14, but not KIF5B, were detected by western blot in protein aggregates upon loss of U$_{34}$-enzymes (Fig. 3e). Of note, KIF4A and KIF14 protein levels were partially rescued after blocking the autophagic flux with chloroquine, but not after proteasome inhibition (Supplementary Fig. 3a). Importantly, the systematic recoding of KIFA4A cDNA (i.e., all AAA, CAA, GAA codons were mutated into their synonymous, U$_{34}$-modification independent, codons AAG, CAG, GAG—KIF4A-Mut, as in refs. [19,20]; Fig. 3f) abolished the translation defects observed upon U$_{34}$-enzymes depletion (Fig. 3g), prevented KIF4A aggregation and restored KIF4A expression levels (Fig. 3h). Of note, no changes in mRNA levels nor in global aggregation propensity were observed upon ELP3 depletion

among cells overexpressing the KIF4A-WT or -Mut constructs (Supplementary Fig. 3b-c). This shows that aggregation of protein targets of the U$_{34}$-tRNA modification pathway (U$_{34}$-targets) is caused by codon-dependent translational defects upon loss of U$_{34}$-enzymes. Taken together, these data indicate that U$_{34}$-enzymes prevent the aggregation and degradation of U$_{34}$-targets through codon-dependent regulation of mRNA translation.

**A penta-hydrophilic amino acid pattern links translation defects to protein aggregation in targets of the U$_{34}$-tRNA modification pathway.** Surprisingly, the above analyses show that defective translation (i.e., ribosome accumulation) in U$_{34}$-enzymes-depleted cells does not systematically correlate with protein aggregation and with subsequent differences in protein expression levels. Indeed, a number of genes are significantly enriched in U$_{34}$-codons but they remain normally expressed at the protein level in cells defective in U$_{34}$-enzymes (Figs. 1e, 2a–c). We surmised that additional features may dictate protein aggregation in the context of defective codon-dependent translation. To identify these features, we performed a series of analyses to identify specific parameters that differentiate proteins that are downregulated versus unchanged upon ELP3 depletion, despite the sharing of high U$_{34}$-codon content

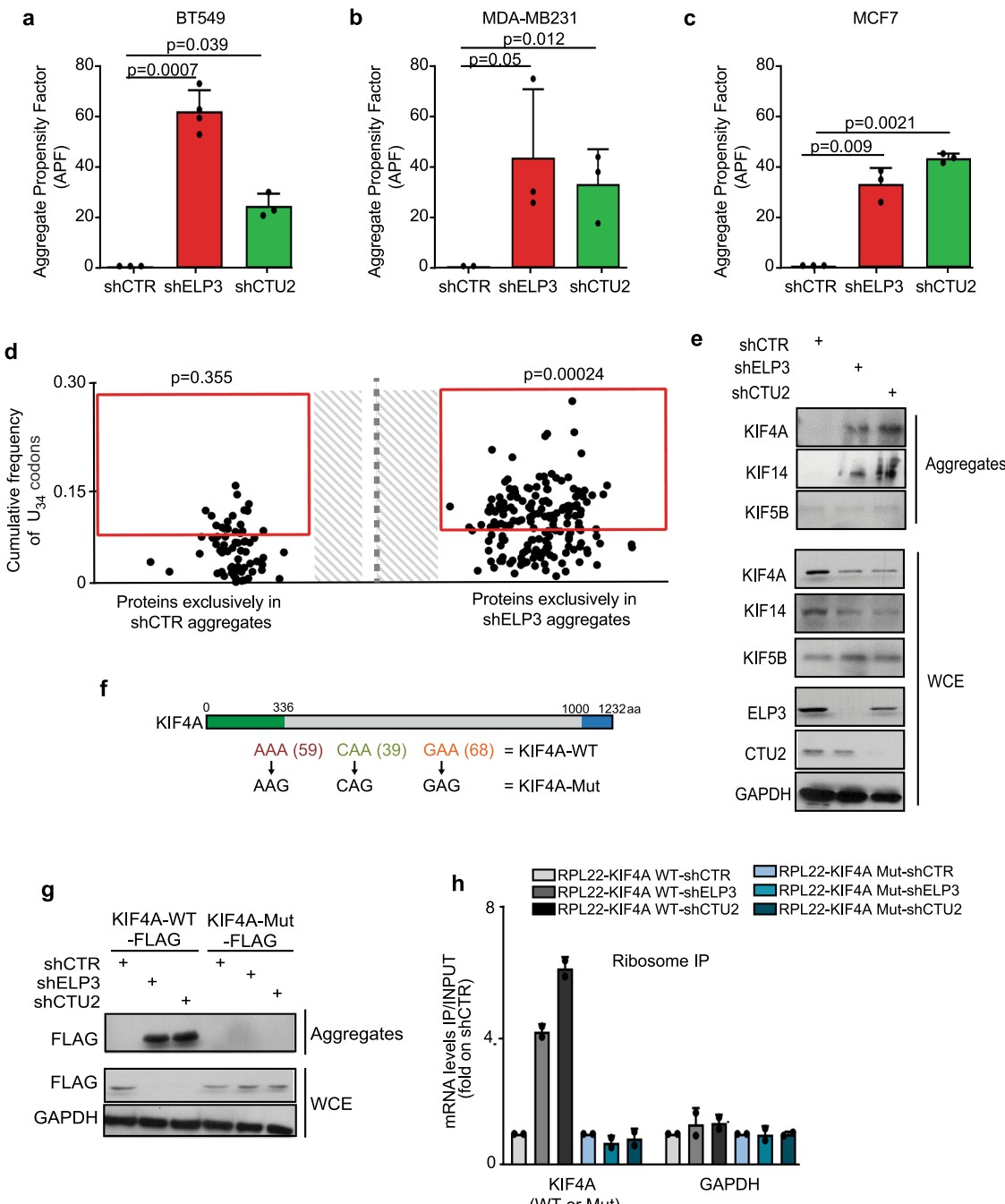

**Fig. 3 Loss of U$_{34}$-enzymes leads to protein aggregation in a codon-dependent manner. a–c** Protein aggregation was measured by FACS in BT549 (**a**), MDA-MD231 (**b**), and MCF7 (**c**) upon loss of ELP3 or CTU2 ($n = 3$ independent experiments, two-tail $t$-test, data are mean + s.d). **d** Protein content of aggregates of MCF7 cells depleted or not of ELP3 were assessed by proteomics ($n = 1$ independent experiment); proteins exclusively found in shELP3 or shCTR conditions are shown. Enrichment of U$_{34}$-codons was assessed by $\chi^2$ test (two-sided). **e** Levels of indicated proteins were determined by western blot in whole-cell extracts (WCE; loaded 30 μg) and aggregates (isolated from >1 mg of protein extract) of MCF7 cells depleted of ELP3 or CTU2 ($n = 2$ replicates). **f** Schematic representation of KIF4A mutant. The kinesin domain (green) and the globular region (blue) of the protein are indicated. **g** Protein levels of whole-cell extracts (WCE) and aggregates of MCF7 cells stably overexpressing FLAG-RPL22 and the identified KIF4A mutants in the presence or absence of ELP3 or CTU2 ($n = 3$ replicates). **h** qRT-PCR after ribosome immunoprecipitation of indicated cells. $n = 2$ independent experiments, data are mean + s.d.

(i.e., AAA, CAA, GAA; $q$ value <0.05; Supplementary Fig. 4a). First, we quantified features linked to mRNAs, namely, expression levels, length, codon content, and U$_{34}$-codons distribution. No significant difference is observed between the two groups (Supplementary Fig. 4b-e). Hence, for each protein of

the two groups, we assessed features linked to protein structure (i.e., number of α-helix and β-sheets), amino acid content, and protein aggregation propensity (calculated by the *Zyggregator* method[34]). Again here, no statistical difference between the two groups is observed (Supplementary Fig. 5a-c). Finally, we

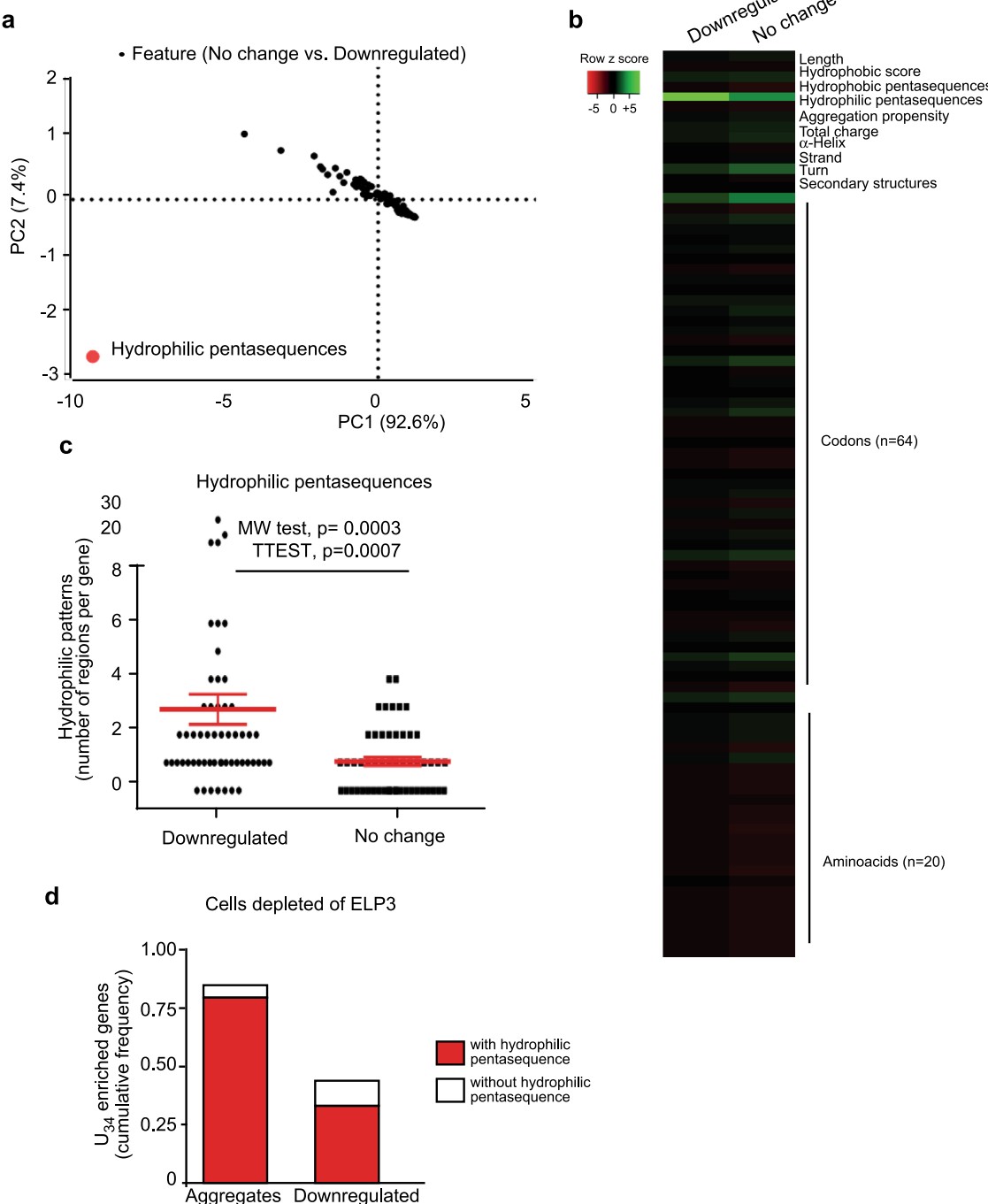

**Fig. 4 A penta-hydrophilic amino acid pattern links translation defects to protein aggregation in U$_{34}$-enzymes depleted cells. a, b** Principal component analysis (**a**) and heat map representation (**b**) of the assessed features in U$_{34}$-enriched (*q* value <0.05) downregulated or unaffected proteins upon depletion of ELP3. **c** Number of hydrophilic pentasequences was assessed in U$_{34}$-enriched (*q* value <0.05) downregulated (*n* = 58) or unaffected proteins (*n* = 56) upon depletion of ELP3. Two-tails *t*-test and Mann–Whitney U (MW) test were performed to assess significance. Data are mean +/− SEM. **d** Presence of hydrophilic pentasequences was assessed in U$_{34}$-enriched (frequency of U$_{34}$-codon >0.088) downregulated proteins (BT549) or in aggregates (MCF7) cells depleted of ELP3.

further evaluated each of the parameters linked to aggregation propensity separately. Strikingly, we found that the presence of patterns of five hydrophilic amino acids significantly segregated proteins that are downregulated upon depletion of ELP3 (*p* = 0.0003; Fig. 4a–c), while total charge, hydrophobic score, and patterns of five hydrophobic amino acids was similar between both groups (Supplementary Fig. 5d-f). In line with these data, proteins harboring hydrophilic pentasequences very

significantly cluster in both ELP3-downregulated proteins as well as in proteins specifically incorporated into aggregates upon ELP3 depletion, indicating a central role of the identified motif in promoting inclusion into aggregates of U$_{34}$-target proteins for degradation (Fig. 4d, Supplementary Data 3). Together, these analyses highlighted that the presence of hydrophilic pentasequences is a key feature to predict protein integration into aggregates upon depletion of U$_{34}$-enzymes.

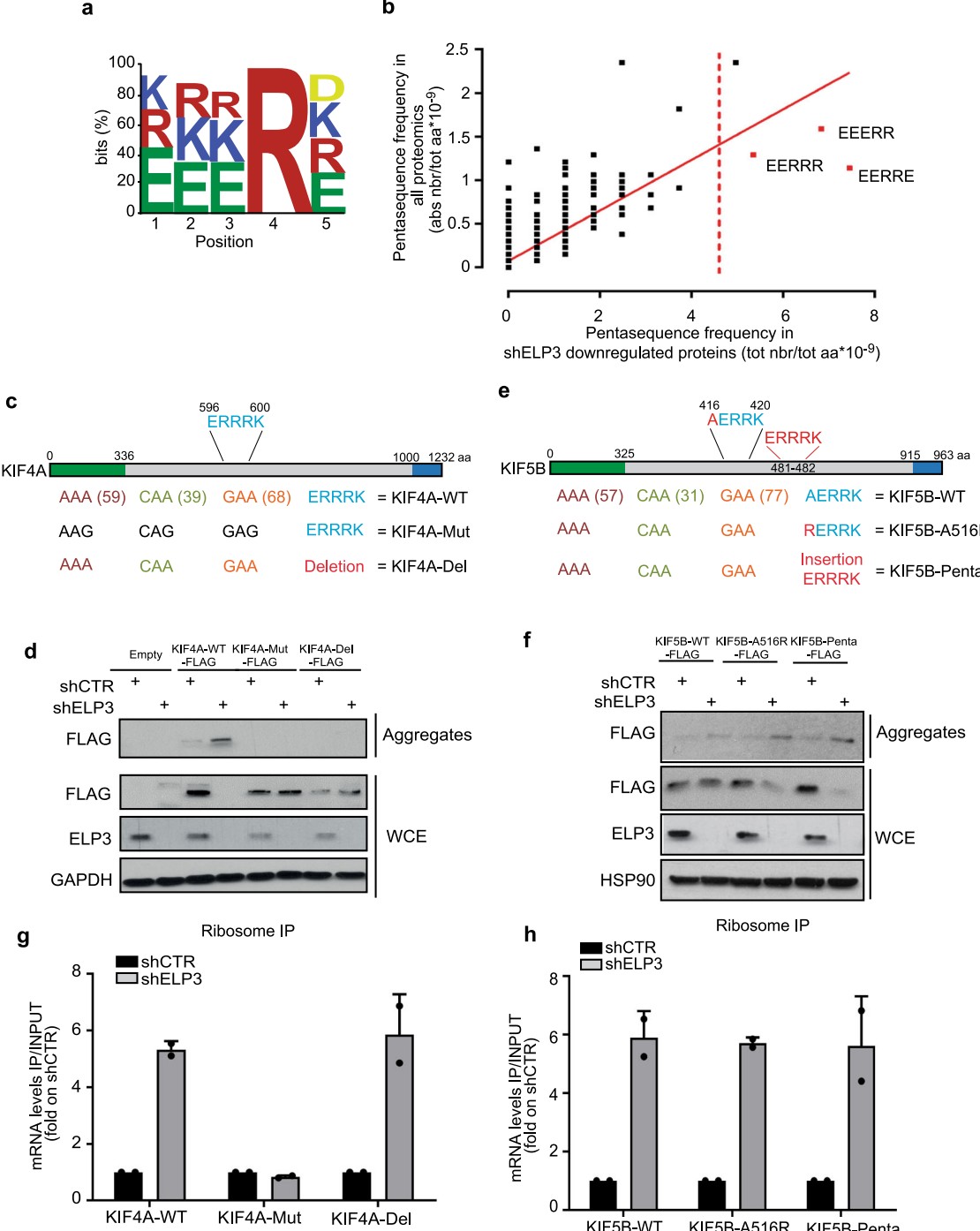

**Fig. 5 Specific hydrophilic motifs dictate protein aggregation upon loss of U₃₄-enzymes.** **a** Identified hydrophilic motif. **b** Assessment of the specific hydrophilic pentasequences (in red) overrepresented in proteins downregulated upon ELP3 depletion. Linear regression is plotted. **c**, **e** Schematic representation of KIF4A (**c**) and KIF5B (**e**) proteins and mutants. Hydrophilic motifs are indicated. The kinesin domain (green) and the globular region (blue) of the proteins are indicated. **d**, **f** Levels of indicated proteins were determined by western blot in whole-cell extract (WCE) and aggregates of MCF7 cells overexpressing the indicated mutants of KIF4A (**d**) and KIF5B (**f**) and depleted of ELP3 (n = 2 replicates). **g**, **h** qRT-PCR after ribosome immunoprecipitation of KIF4A (**g**) and KIF5B (**h**) mutants. n = 2 independent experiments, data are mean + s.d.

**Specific hydrophilic motifs dictate protein aggregation upon loss of U₃₄-enzymes.** Having uncovered the importance of hydrophilic pentasequence patterns in the definition of protein targets of the U₃₄-tRNA modification pathway, we combined all identified pentasequences and highlighted a consensus motif [EKR]-[EKR]-[EKR]-R-[DEKR] (Fig. 5a). This motif is present in 10.16% of the protein-coding genes (Supplementary Fig. 6a) and is strongly associated with the frequency of U₃₄-codons in genes

($\chi^2$ test: $p = 9.84 \times 10^{-48}$; Supplementary Data 4). Furthermore, the motif discriminates proteins that are downregulated ($\chi^2$ test: $p = 3.88 \times 10^{-18}$; Fig. 5b) and directed into aggregates upon ELP3 depletion ($\chi^2$ test: $p = 8.4 \times 10^{-8}$; Supplementary Data 4). Finally, the combination of the two features, the U₃₄-codon content and the presence of hydrophilic motifs, better predicts protein expression levels downstream of the U₃₄-tRNA modification pathway ($\chi^2$ test downregulated proteins vs. no change: $p = 6.58 \times 10^{-10}$;

Supplementary Data 4). Of note, we did not find any correlation between the presence and the distribution of the motifs and protein length (Supplementary Fig. 6b-c). Finally, we computed the conservation scores of the identified motif (Supplementary Fig. 6d) against random sequences of five contiguous amino acids across 10 mammalian species (Supplementary Fig. 6e). Interestingly, we show that the motif is conserved ($p = 7.05 \times 10^{-20}$) suggesting a selection pressure during evolution.

To further assess the biological relevance of the identified motif in determining protein fate upon $U_{34}$-tRNA modification-dependent translation elongation defects, we took advantage of two kinesin proteins whose mRNAs are both highly enriched in $U_{34}$-codons: (1) KIF4A, which harbors a consensus motif (i.e., ERRRK; Fig. 5c), and is detected in protein aggregates upon depletion of $U_{34}$-enzymes (Fig. 5d); and (2) KIF5B, which does not contain any consensus motif (Fig. 5e), is not downregulated and therefore not detected in protein aggregates upon depletion of $U_{34}$-enzymes (Fig. 5f). We deleted the hydrophilic motif in the KIF4A sequence and inserted one in the KIF5B sequence and we assessed protein expression of the constructs upon depletion of $U_{34}$-enzymes. Strikingly, the deletion of the ERRRK motif in KIF4A (KIF4A-Del; Fig. 5c) is sufficient to rescue its expression level and to prevent KIF4A aggregation in the absence of ELP3 (Fig. 5d). This effect does not rely on changes in mRNA translation elongation (i.e., ribosome occupancy), which remains unaffected by the absence of the motif (Fig. 5g and Supplementary Fig. 7a). Conversely, the addition of the ERRRK motif in the KIF5B cDNA sequence (either by a single amino acid permutation—KIF5B-A516R—or by insertion of the whole ERRRK sequence—KIF5B-Penta; Fig. 5e) was sufficient to promote KIF5B protein aggregation and downregulation upon ELP3 depletion (Fig. 5f). Again, the addition of the motif does not affect KIF5B mRNA translation elongation (Fig. 5h and Supplementary Fig. 7b). These data indicate that the presence of the hydrophilic motif is necessary to promote protein aggregation and downregulation of genes that undergo translation elongation defects upon depletion of $U_{34}$-enzymes.

We then wanted to assess if the presence of a hydrophilic motif could discriminate the proteins whose downregulation is directly dependent on the loss of the $U_{34}$-tRNA modification (i.e., proteins incorporated in aggregates in a codon-dependent manner) from those that are indirectly downregulated in these conditions. To this aim, we selected two proteins that share similar $U_{34}$-codons frequency, are downregulated upon ELP3 depletion and are part of the same kinetochore complex[35]. We generated $U_{34}$-codon mutants cDNA of NCD80 and NUF2 (as before; Fig. 6a) and we assessed their expression levels upon $U_{34}$-enzymes depletion. Surprisingly, even though NCD80 and NUF2 are in the same protein complex, share a similar frequency of $U_{34}$-codons and display comparable ribosome accumulation at their mRNA upon $U_{34}$-enzymes loss (Supplementary Fig. 7c), the systematic recoding of $U_{34}$-codons by their synonymous only rescued NCD80 expression (Fig. 6b) but did not impact NUF2 levels (Fig. 6c). Importantly, only NCD80 harbors hydrophilic motifs in its sequence (Fig. 6a) and is therefore detected in protein aggregates upon ELP3 depletion (Fig. 6d). Conversely, NUF2 sequence does not contain any hydrophilic motif and it is never detected in protein aggregates upon ELP3 depletion (Fig. 6d). Hence, NUF2 downregulation is partially reverted upon expression of the NDC80-Mut construct (Fig. 6e). This indicates that NUF2 downregulation likely results from the loss of its partner NDC80 and not from an intrinsic dependence on the $U_{34}$-tRNA modification pathway. These data reveal that the presence of the hydrophilic motif discriminates between direct (i.e., NDC80) and indirect (i.e., NUF2) targets of the $U_{34}$-tRNA modification pathway. All together this set of data demonstrates that the presence of hydrophilic motifs is necessary to mediate

protein aggregation and clearance of $U_{34}$-targets upon loss of $U_{34}$-enzymes.

**The charged/ampholytic nature of the hydrophilic motif mediates protein aggregation upon depletion of $U_{34}$-enzymes.** The identified hydrophilic pentamer is defined by its unique amino acid composition, exclusively enclosing charged residues, but also presenting an ampholytic nature (i.e., has the characteristics of both an acid and a base). We surmised that this unique feature might be the reason why proteins are subjected to aggregation upon depletion of $U_{34}$-enzymes. In line with this hypothesis, we tested two non-exclusive models that may explain the role of the hydrophilic motif in protein aggregation. First, the ampholytic nature of the motif may drive interaction with the ribosome exit tunnel during translation elongation. The ribosome exit tunnel is a negatively charged cavity from which the nascent proteins arise[36]. Different factors are involved in facilitating or opposing the exit of the nascent peptide from the ribosome tunnel, among these, tRNA availability, protein secondary structure formation and the presence of charged amino acids in the nascent peptide[37–39]. We recently computed the ribosome exit tunnel electrostatics in a model to quantify the energetic needs required to effectively exit nascent peptides from the ribosome tunnel[40]. Using this model, we found that the presence of the hydrophilic motif in the KIF4A protein sequence represents a region of high energetic needs in order to exit the ribosome tunnel and to progress toward translation elongation (Fig. 7a). Conversely, the deletion of the hydrophilic motif in the KIF4A protein sequence (i.e., in KIF4-Del), which prevents KIF4A aggregation upon depletion of $U_{34}$-enzymes, alters the local energetic barrier, which is predicted to favor ribosome movement across this region (Fig. 7b). This observation indicates that the presence of the hydrophilic motif brings an additional energetic barrier during translation elongation, which may be the cause of protein misfolding and aggregation.

Second, recent evidences describe the propensity of low complexity regions (LCR) rich in charged amino acids to regulate protein phase behavior, to promote liquid–liquid phase separation (LLPS) and to trigger protein aggregation[41–45]. Therefore, we speculated that the ampholytic nature of the hydrophilic motif might trigger LLPS and auto-aggregation among the $U_{34}$-target proteins, upon $U_{34}$-enzymes depletion. We visualized KIF4A-WT or KIF4A-Del (i.e., that does not harbor the hydrophilic motif) proteins in MCF7 cells by immunofluorescence. Interestingly, upon depletion of $U_{34}$-enzymes, KIF4A-WT aggregates in small punctuated structures, similar to SHP2 mutants-LLPS associated structures[46] (Fig. 7c). Strikingly, KIF4A-Del is never found in these structures in $U_{34}$-enzymes depleted cells (Fig. 7c). We also assessed the amyloid aggregation propensity in KIF4A-WT and KIF4A-Del using the TANGO algorithm[47]. Interestingly, the hydrophilic motif we identified in KIF4A is predicted to have high propensity to undergo amyloid aggregation (Fig. 7d). Moreover, its deletion (i.e., in KIF4A-Del) causes a net reduction in amyloid aggregation propensity in KIF4A, as measured by the total area under the curve (KIF4A-WT = 163231 and KIF4A-Del = 146070), and a change in the maximal amyloid aggregation peak (KIF4A-WT max peak—intensity 34359—is found in bin 9 that harbors the motif, while KIF4A-Del max peak—intensity 24352—is found in bin 14) (Fig. 7d). Conversely, deletion of the motif did not affect KIF4A general aggregation propensity (Supplementary Fig. 8a) nor its presence was found associated with aggregation propensity calculated by the Zyggregator method (Supplementary Fig. 8b). This indicates that the presence of the charged/ampholytic hydrophilic motif may trigger

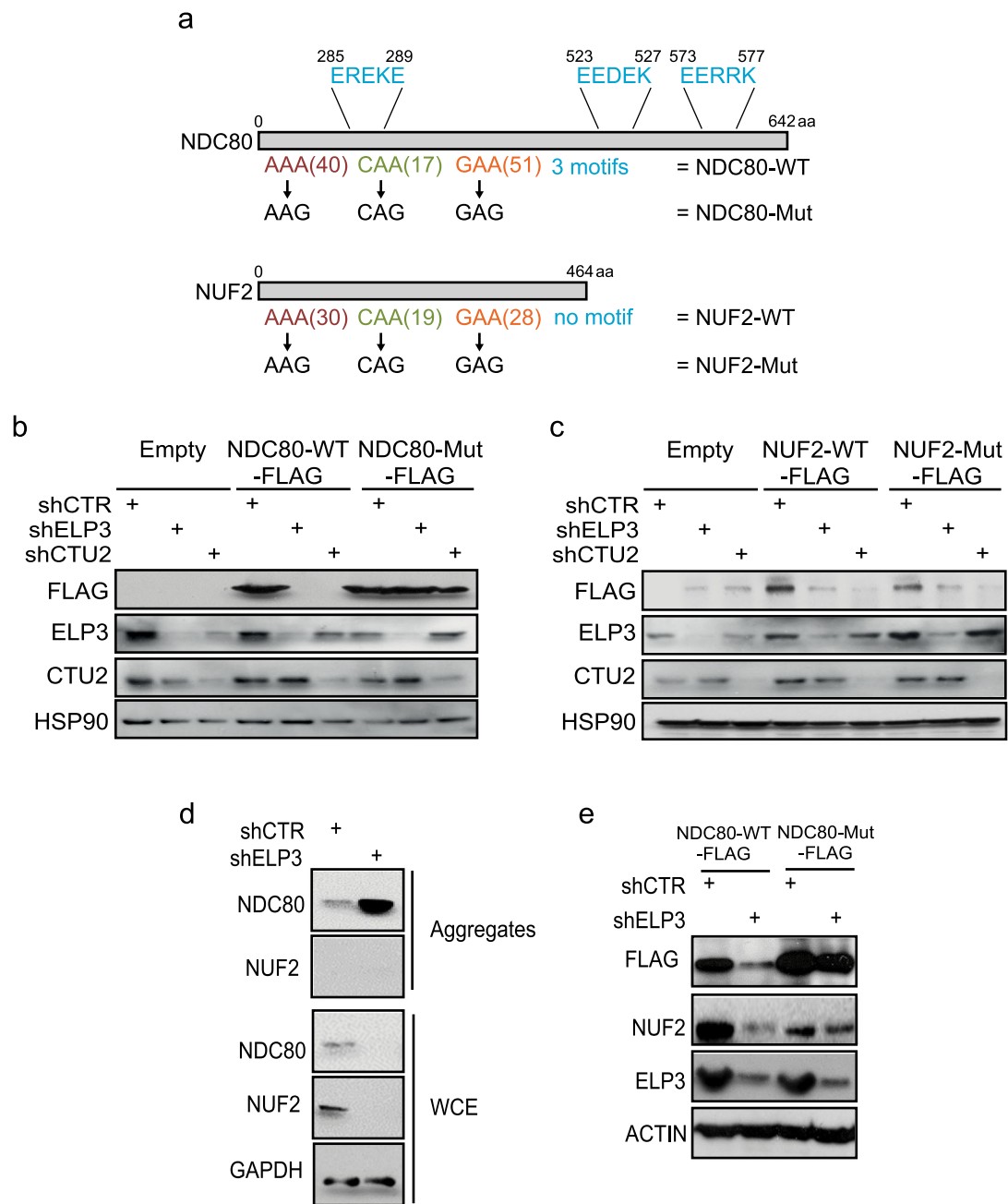

**Fig. 6 The presence of hydrophilic motifs discriminates between direct and indirect targets of the $U_{34}$-tRNA modification pathway. a** Schematic representation of the NDC80 and NUF2 protein sequences. Hydrophilic motifs are indicated in blue. **b** Protein levels of indicated proteins in cells overexpressing control or NDC80-WT-FLAG or NDC80-Mut-FLAG constructs in the presence or absence of ELP3 or CTU2 ($n = 2$ replicates). **c** Protein levels of indicated proteins in cells overexpressing control or NUF2-WT-FLAG or NUF2-Mut-FLAG constructs in the presence or absence of ELP3 or CTU2 ($n = 2$ replicates). **d** Levels of indicated proteins were determined by western blot in whole-cell extracts (WCE) and aggregates of MCF7 cells depleted of ELP3 ($n = 2$ replicates). **e** Levels of indicated proteins were determined by western blot in cells overexpressing NDC80-WT-FLAG or NDC80-Mut-FLAG constructs in the presence or absence of ELP3 ($n = 1$ replicate).

aggregation through changes in protein phase behavior and LLPS upon depletion of $U_{34}$-enzymes.

## Discussion
Recently, the role of tRNA biology in health and disease has gained an increasing amount of attention[11,48,49]. The development of ribosome sequencing techniques allowed to assess the exact role of tRNA modifications, such as $mcm^5s^2$ $U_{34}$-tRNA modification, in translation regulation and specific codon decoding[16,17,19]. Currently, it is assumed that impairments in

ribosome kinetics at mRNAs are directly reflected in changes in expression of the corresponding proteins. Nevertheless, very few studies focus on the mechanisms linking codon-specific ribosome pausing and protein homeostasis. Specifically, how perturbation of codon-dependent-translation impacts subsequent protein expression output remains poorly understood. Surprisingly, here we show that codon-dependent translation elongation defects caused by the depletion of $U_{34}$-enzymes are not systematically associated with changes in protein expression. While translation elongation defects upon perturbation of $U_{34}$-tRNA modification

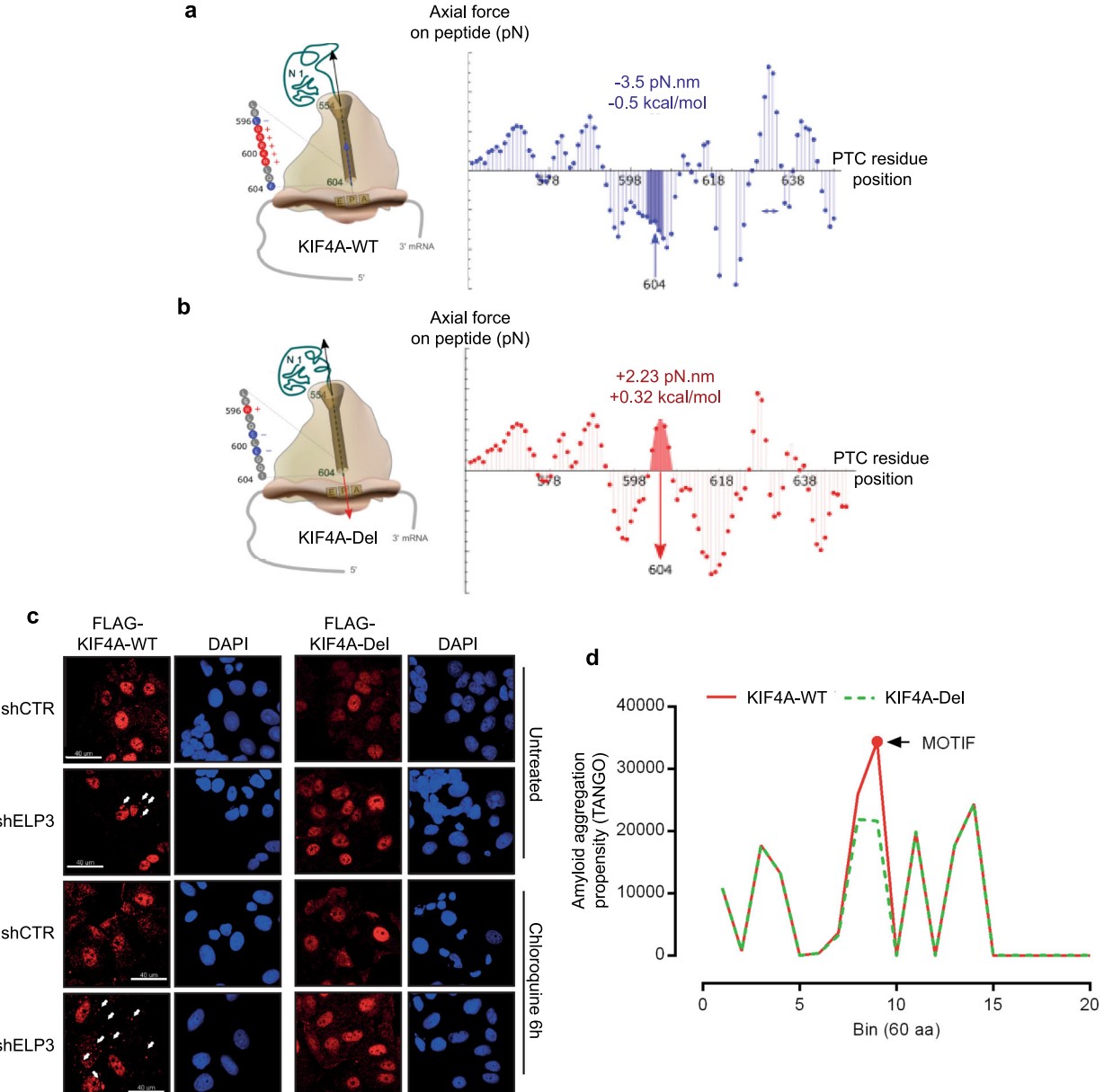

**Fig. 7 The ampholytic nature of the motif drives protein aggregation upon depletion of $U_{34}$-enzymes. a, b** Axial forces profile on nascent chain caused by the electrostatic interaction in the ribosome exit tunnel (**a**) wild-type KIF4A; (**b**) ERRRK deleted KIF4A. The axial forces profiles and local mechanical work acting on the two chains are different upon incorporation of residues 596–646 at the peptidyl-transferase center (PTC). Schematic representation of ribosome–protein interaction, on the left. **c** Anti-Flag immunofluorescence of MCF7 cells depleted or not of ELP3 and overexpressing KIF4A-WT or KIF4A-Del in control or after 6 h of treatment with 100 μM chloroquine. Arrows depict KIF4A aggregates ($n = 2$ replicates). **d** Amyloid aggregation profiles of KIF4A-WT and KIF4A-Del calculated by using the TANGO algorithm.

are strictly dependent on codon content, the consequences of these defects on protein output are determined by other features. We identify hydrophilic amino acid motifs as essential to dictate protein aggregation and subsequent degradation in contexts of defective translation elongation at $U_{34}$-codons. Therefore, we propose a model in which codon-dependent regulation of translation elongation by wobble tRNA modification and maintenance of protein integrity are two independent processes (see the model in Fig. 8). Importantly, the previously identified protein targets of the $U_{34}$-tRNA modification pathway (i.e., SOX9[25], DEK[20], HIF1α[19], and hnRNPQ[26]; Fig. 1c–e) all harbor at least one hydrophilic motif in their sequences, supporting the model here proposed (Supplementary Fig. 8c). Of note, a portion of genes enriched in $U_{34}$-codons and downregulated

upon depletion of $U_{34}$-enzymes do not harbor hydrophilic motifs and are not directed toward aggregation (for instance NUF2; see Supplementary Table 3). The mechanism leading to their downregulation upon depletion of $U_{34}$-enzymes remains to be understood.

How the presence of hydrophilic motifs mediate protein aggregation remains to be precisely established. The motif strongly differs from known chaperone- and cochaperone-binding sites, which are generally hydrophobic[50]. In contrast, the here identified motif shares some similarities with the motifs responsible for chaperone-mediated autophagy (CMA) (i.e., KFERQ and derivatives)[51]. Nevertheless, the unbiased search for CMA motifs[51] showed no overlap with the hydrophilic motifs identified herein, making this mechanism of degradation unlikely

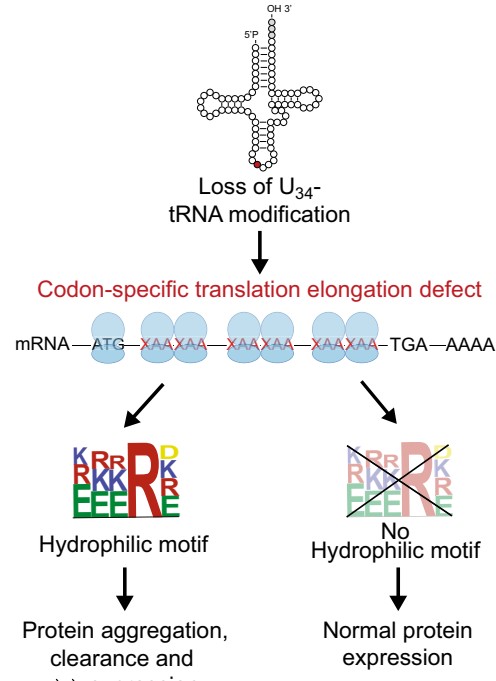

**Fig. 8 Proposed model.** Loss of $U_{34}$-enzymes invariably determines pausing and accumulation of ribosomes on transcripts rich in XAA codons. The result of this pausing on protein expression is determined by the presence of hydrophilic motifs that mediate protein aggregation and subsequent clearance. See text for details.

(Supplementary Fig. 8d-e). Previous studies highlighted the potential role of hydrophilic stretches as gatekeepers for the correct folding of nascent proteins due to their proximity to aggregation-prone regions[52–55]. Although, we do not exclude this possibility, in the KIF4A protein sequence, the hydrophilic motif sits in the proximity of low aggregation-prone regions. Rather, our data show that the unique amino acid composition of the motif may underlie protein aggregation upon $U_{34}$-enzymes depletion. Indeed, we show that the charged and ampholytic nature of the motif may affect phase behavior of the $U_{34}$-protein targets, possibly triggering LLPS and consequently promoting protein aggregation upon depletion of $U_{34}$-enzymes. For example, the here identified motif shares physical characteristics with fragments in the low complexity (LC) domains in the U1-70K protein, that have been shown to drive LLPS and aggregation in cells[41]. Also, arginine-rich domains mediate LLPS via electrostatic interactions, which therefore requires multiple proteins or motifs. Why would the perturbation of $U_{34}$-tRNA modification and the subsequent codon-dependent deregulation of translation favor this mechanism is not known. It is possible that the translational defects seen upon depletion of $U_{34}$-enzymes might result in protein downregulation, only if the exposure of charged/ampholytic motifs is sufficient to trigger LLPS-driven aggregation that will lead to clearance by the lysosome.

The charged/ampholytic nature of the pentamer could trigger aggregation also by its interaction with the ribosome exit channel tunnel during translation elongation. Previous studies demonstrated that factors involved in facilitating or opposing the exit of the nascent peptide from the ribosome tunnel can impact on protein folding[56]. Interestingly, in KIF4A, the identified hydrophilic motif represents a region of high energetic needs in order to exit the ribosome tunnel and to progress toward translation elongation (Fig. 7c). Conversely, its deletion, which prevents incorporation of KIF4A in aggregates upon

depletion of the $U_{34}$-enzymes (Fig. 5d), alters the local energetic barrier, likely favoring ribosome movement across this region (Fig. 7d). It is therefore tempting to speculate that the additional energetic barrier brought by the presence of the hydrophilic motif could lead to an excessive ribosome pausing and may be the cause of protein misfolding and aggregation. Further work is required to further clarify these mechanisms.

Our data open the possibility that the uncoupling of translation elongation defects and maintenance of protein homeostasis may generally apply to other tRNA modification pathways. Understanding the mechanisms linking codon-dependent translation defects to maintenance of protein expression for all the still poorly studied tRNA modification pathways will help define the importance of this layer of regulation in proteome expression.

## Methods
**Cell lines and reagents.** Breast cancer cell lines (BT549, MDA-MB231, and MCF7) were from ATCC and were grown in Dulbecco's modified Eagle's medium (Gibco) containing 10% of fetal bovine serum, 1% glutamine, and 1% penicillin–streptomycin. Lines were tested for mycoplasma contamination on a regular basis.

The antibodies, primers, and shRNA sequences used in this study are listed in Supplementary Data 5.

**Western blot and immunoprecipitation.** Cells were lysed in lysis buffer (50 mM Tris-HCl pH 7.5, containing 2 mM EDTA, 150 mM sodium chloride, 0.5% sodium deoxycholate, 50 mM sodium fluoride, 1% NP-40, and 0.1% SDS, Complete™ proteinase inhibitor cocktail (Roche)) and protein determination was assessed using BCA Protein Assay Kit from ThermoFisher and according to manufacturer's instructions.

**qPCR.** Total RNA was extracted using the E.Z.N.A. Total RNA Kit I (OMEGA Bio-Tek) according to the manufacturer's instructions. Subsequently, 1 μg of total RNA was used to synthetize the corresponding cDNA using the RevertAid H Minus First Strand cDNA Synthesis Kit (ThermoFisher). Quantitative evaluation of selected genes was determined by SYBR-Green-based qPCR performed using the Light Cycler 480 (Roche). All determinations were performed in triplicates. The relative expression of the target gene transcript and reference gene transcript was calculated as ΔΔCt.

**Viral infections.** For stable lentiviral shRNA infections, all constructs were purchased from Sigma. The kinesins FLAG-tag overexpression plasmids (KIF4A, KIF5B, and mutants) were generated from the pLV-Puro-hPGK plasmid by Vectorbuilder. NUF2 and NDC80 FLAG-tag overexpression plasmids and codon mutants were generated from the Lego-Ig2 plasmid by Genscript. The Luciferase/Nanoluciferase constructs were derived from pLV-Puro-hPGK and generated by Vectorbuilder (maps are available upon request). Lentiviral infections were carried out as previously described[57]. Briefly, in all cases, HEK293-LentiX producer cells were transfected with 12 μg of lentiviral vector, 5 μg VSVG, and 12 μg R.891 and Mirus (3:1) for 48 h. Supernatant containing virus was filtrated using a 45 μm filter and polybrene was added to a final concentration of 8 μg/ml, and virus were added to target cells. Cells were selected for 3 days with 1 μg/ml puromycin. Knockdown or overexpression efficiency was evaluated by RT-qPCR and/or by western blot analysis.

**Ribosome immunoprecipitation.** Ribosome pulldown was performed as previously described[19]. Briefly, cells were stably transduced with RPL22-FLAG expressing lentiviral construct and treated for 15 min with 100 μg/ml cycloheximide. Then, cells were lysed in 20 mM HEPES-KOH (pH 7.3), 150 mM KCl, 10 mM MgCl₂, 1% NP-40, EDTA-free protease inhibitors, 0.5 mM DTT, 100 μg/ml cycloheximide, and 10 μl/ml rRNasin and Superasin in RNase-free water, and incubated overnight with anti-FLAG beads. After incubation, beads were washed five times with a high salt buffer (350 mM KCl) and three times with a low salt buffer (150 mM KCl). Beads were then washed in RNA extracting buffer with β-mercaptoethanol as previously described and underwent RT-qPCR.

**Immunofluorescence.** 200000 MCF7 cells depleted or not of ELP3 were seeded and treated as indicated. Cells were fixed in 4% PAF and permeabilized in PBS supplemented with Triton 0.2%. Primary antibody was kept overnight at 4 °C in PBS supplemented with 0.1% TRITON and 5% BSA (dilution: 1/200). Appropriate secondary antibody (dilution: 1/1000) and DAPI were added for 1 h at room temperature in PBS supplemented with TRITON 0.1% and BSA 3%. Images were acquired by confocal microscopy (confocal Leica SP5).

**RNA sequencing**. BT549 cells were stably depleted of ELP3 by shRNA (see the "Viral infections" section); a scramble shRNA was used as control. Total RNAs from three independent infections were extracted using TRIzol (Life Technologies), according to the manufacturer's protocol. RNA integrity was verified on the Bioanalyser 2100 with RNA 6000 Nano chips, and RIN scores were >9 for all samples. The Illumina Truseq RNA Sample Preparation kit V2 was used to prepare libraries from 500 ng total RNAs. PolyA RNAs were purified with polyT-coated magnetic beads, chemically fragmented, and used as template for cDNA synthesis using random hexamers. cDNA ends were subsequently end-blunted, adenylated at 3'OH extremities, and ligated to indexed adaptors. Finally, the adapters ligated library fragments were enriched by PCR following Illumina's protocol and purified. Libraries were validated on the Bioanalyser DNA 1000 chip and quantified by qPCR using the KAPA library quantification kit. Sequencing was performed on HiSeq2000 in paired-end $2 \times 100$ base protocol. For data analysis, the nf-core-rnaseq pipeline was used.

**Proteomic study**. BT549 cells depleted or not of ELP3 by shRNA were lysed in SDS 2% buffer containing 50 mM Tris-HCl pH 8, 150 mM sodium chloride, 10 mM sodium fluoride, and 1 mM trisodium phosphate, Complete™ proteinase inhibitor cocktail (Roche). At least 1 mg of each sample was sent for mass spectrometry analysis using LC/MS/MS (MsBioworks, LLC). Samples were TCA precipitated overnight at −20 °C in 90/10 acetonitrile/water incorporating 20% TCA. Precipitates were washed, pelleted, and resuspended in 5 M urea, 50 mM Tris-HCl, pH 8.0, Complete™ proteinase inhibitor cocktail (Roche) and quantitated by Qubit fluorometry (Life Technologies). A 50 μg aliquot per sample was reduced using dithiothreitol, alkylated with iodoacetamide, and then subjected to digestion with trypsin (Promega) for 18 h at 37 °C at an enzyme-to-substrate ratio of 1:50. The digestion was quenched with formic acid and peptides cleaned using solid-phase extraction (SPE) using the Empore C18 plate (3 M). A 2 μg aliquot of each sample was analyzed by nano LC/MS/MS with a Waters M-Class HPLC system interfaced to a Fusion Lumos tandem mass spectrometer (ThermoFisher). Peptides were loaded on a trapping column and eluted over a 75 μm analytical column at 350 nL/min; both columns were packed with Luna C18 resin (Phenomenex). A 4 h gradient was employed. The mass spectrometer was operated in data-dependent mode, with MS and MS/MS performed in the Orbitrap at 60,000 FWHM resolution and 15,000 FWHM resolution, respectively. APD was turned on. The instrument was run with a 3-s cycle for MS and MS/MS. The experiment was done in triplicate.

Thermo RAW files were processed using MaxQuant v1.6.2.3 (Max Planck Institute). Production data were searched against the combined forward and reverse Swissprot H. sapiens protein database using Andromeda. Data were filtered 1% protein and peptide-spectrum match (PSM) level false discovery rate (FDR) and requiring at least one unique peptide per protein. Label-free protein quantitation was based on the extracted ion chromatograms of detected peptides; data were normalized using the embedded LFQ algorithm.

**Protein aggregates determination**. Protein aggregates formation was monitored by FACS using PROTEOSTAT® Aggresome Detection Kit (ENZO) according to manufacturer instructions and as previously described[19]. Fluorescence was measured by FACS by FL3 channel. Aggresome propensity factor (APF) was calculated using the formula: 100*(mean fluorescence intensity depleted − mean fluorescence intensity control)/mean fluorescence intensity depleted.

Isolation of protein aggregates was performed as previously described[19]. Briefly, cells were pelleted, snap-frozen in liquid nitrogen, and resuspended in lysis buffer (20 mM Na-phosphate, pH 6.8, 10 mM DTT, 1 mM EDTA, 0.1% Tween, 1 mM PMSF, protease inhibitor cocktail [Roche] and DNAse I). Lysates were kept 20 min at room temperature and sonicated. After centrifugation at 200 × g for 20 min at 4 °C, supernatants were collected and protein dosage performed. Protein extracts were then centrifuged for 20 min 16,000 × g at 4 °C. Pellets were washed (20 mM Na-phosphate, pH 6.8, 1 mM PMSF, protease inhibitor cocktail, 2% NP-40) and sonicated. Final pellets were washed with buffer with no NP-40 and sonicated again. For protein detection by western blot, pellets were resuspended in 1% SDS.

For proteomics analysis, aggregates were purified from the cytoplasmic fraction after cyto-nuclear extraction. Briefly, cells were lysate with cytoplasmic lysis buffer (1 M Tris-HCl pH 7.9, 0.2 CaCl₂, 0.25 M EDTA, 2 M MgCl₂, protease inhibitors) and supernatant was collected after centrifugation (4 °C for 10 min at 17,000 × g). After aggregates isolation, pellets were washed twice in 1× PBS, and analyzed by LC/MS/MS.

**Luminescence**. Luminescence of indicated stably transduced MDA-MB231 cell lines was measured using Promega Dual-Luciferase® Reporter Assay according to manufacturer instructions. Data were expressed as fold of nanoluc/firefly ratio.

**Computational analysis of gene-specific codon usage (GSCU) patterns**. We downloaded 23,099 homo sapiens gene sequences from the human genome database (HGD), and then analyzed each gene using our GSCU algorithm. The GSCU algorithm interrogated a gene from the start to the stop codon. The frequency of a specific codon (i.e., AAA, GAA, and CAA) used in a gene was counted and further compared with the codon frequency in another gene, which was randomly picked up. 10,000 times comparison was done. By such comparison we got the p-value,

which means the possibility that the randomly picked up gene has higher frequency of that specific codon than the interest one. We then used q package with the R program, and we generated the $q$ value for each gene. Using $q = 0.05$ as cut off, we picked up the genes enriched for the $U_{34}$-codons AAA, GAA, and CAA.

**Codon enrichment analysis**. Genes clustering was performed by filtering for statistical increase in specific codons content ($q < 0.05$). Genes with AAA, CAA, and GAA $q$ value <0.05 subsequently underwent Toppgene analysis in order to uncover enriched biological process and gene families.

**Statistical analysis**. Gene set enrichment analysis was performed using Toppfun software and GO_biological process database. A FDR $q$ value lower than 0.05 was taken into account. Bubble graphs were created in excel: distance between the clusters was calculated as the frequency of overlapping genes between clusters. The first 30 significant pathways were considered.

Distribution of $U_{34}$-codons across mRNAs was assessed by computing the $U_{34}$-codons present in genes. To this end, we divided into 100 bins the 116 genes of interest and calculated the frequency of the $U_{34}$ codons for each bin. Kolmogorov–Smirnov test was performed to assess significance.

Principal component analyses were performed using the FactoMineR and Factoextra R packages. The script is available upon request. Chi-square test was performed to assess significance.

Aggregation propensity was evaluated by two methods: (1) the Zyggregator method[34], (2) the TANGO algorithm[47]. Aggregation by the zyggregator method was calculated by using the $\log(k) = \alpha_0 + \alpha^{hydr} + \alpha^{ss} + \alpha^{ch} + \alpha^{pat}$ formula; where $\alpha_0$ indicates pH and temperature (here set to 0), $\alpha^{hydr}$ indicates the Hydrophobic score calculated as the sum of the hydrophobic value from ref. [58]. $\alpha^{ss}$ indicated the absolute number of secondary structures such as α-helix and β-sheets according to Uniprot database, $\alpha^{ch}$ indicates the total charge of each protein calculated by adding the net charge of +1 or −1 for positively or negatively charged amino acids, respectively. $\alpha^{pat}$ indicates a pattern for five consecutive hydrophobic or hydrophilic residues in the sequence; a value of +1 was given to each pentasequence. Aggregation calculated by the TANGO algorithm was performed assuming the following parameters: pH = 7; temperature = 298.15 K; ionic strength is 0.02 M; concentration = 1 M; no protection of N and C-term. Both beta-sheet aggregation and amyloid aggregation were assessed.

Analysis of the chaperone-mediated autophagy motifs was done by the KFERQ-finder software[51].

The mapping of the electrostatic potential within the ribosomal exit tunnel and the energy needed by the nascent peptide to exit the tunnel was performed as in ref. [40].

**Motif search**. Hydrophilic pentasequences ($n = 161$) were aligned by ClustaIW in UNIGE to find a consensus motif. Motif search ([EKR]-[EKR]-[EKR]-R-[DEKR]) in different datasets was performed by Motif search tool by Genome.net.

For the conservation analysis, 1466 multiple-alignments were conducted for each subset of 10 orthologs. The human protein was used as source for all the 9 target mammalian orthologs in order to be aligned using the CLUSTAL-W algorithm in Bio-Python. Conservation scores for all the hydrophilic pentasequences and for random sequence of 5 contiguous residues were calculated (Shannon entropy conservation score).

**Reporting summary**. Further information on research design is available in the Nature Research Reporting Summary linked to this article.

## Data availability

Proteomics data are available at ProteomeXchange Consortium (PXD019590; DOI: 10.6019/PXD019590 and PXD019620; DOI: 10.6019/PXD019620) and RNA-seq data are available at ArrayExpress under accession code E-MTAB-9206. Open access databases used in this work include Swissprot (H. sapiens protein database, https://www.uniprot.org/uniprot/?query=*&fil=organism%3A%22Homo+sapiens+%28Human%29+%5B9606%5D%22+AND+reviewed%3Ayes); Human genome database (https://www.ncbi.nlm.nih.gov/grc/human), GO_biological process database (http://geneontology.org/). The data supporting the findings of this study are available from the corresponding authors upon reasonable request. Source data are provided with this paper.

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

## Acknowledgements

We are grateful to the GIGA-proteomics, imaging, genomics, and viral vector facilities for their assistance. This study was supported by an Incentive Grant for Scientific Research (MIS F:4532.13) from the FNRS, the Concerted Research Action Program (tRAME), and Special Research Funds (FSR: C-15/44) at the University of Liege, the Belgian Foundation against Cancer (F/2016/840 and F/2019/097), as well as by the Walloon Excellence in Life Sciences and Biotechnology (WELBIO). We also thank the National Natural Science Foundation of China (NSFC, No. 81803581) and Shanghai Institutions of Higher Learning-2018 (TP2018080, JZ2018005). We are also grateful to the "Foundation Leon Fredericq" for its financial support. N.E.H. is Research Fellow, F.R. and C.D. are Research Associate, A.C. is Research Director, and P.C. is Senior Research Associate at the FNRS, respectively.

## Author contributions

F.R. and P.C. designed the study and analyzed experimental data; F.R., Z.Z., G.V., P.L., M.J., M.G., and P.C. performed software and bioinformatic analyses; F.R., Z.Z., A.M.R.S., N.E.H., S.L., and K.S. performed experiments; G.M. and D.B. performed proteomics, P.C. provided funding. A.C., G.H., C.D., L.G., and P.C. provided resources; F.R. and P.C. wrote the paper. All authors discussed the results and commented on the manuscript.

## Competing interests

A.C., P.C., and F.R. are named on a Belgian patent application (BE2021/5237) related to the cell-based method described in this manuscript. The other authors declare no competing interests.
