## [Peer Review File · Nature Communications]

REVIEWER COMMENTS

Reviewer #1 (Remarks to the Author):

It is established that changes in translation elongation speed caused by depletion of U34 tRNA modifying enzymes elicit a widespread protein aggregation in vivo. Rapino et al. here take a step ahead. First, they observe by combining RNA-seq and proteome analyses that an enrichment in codons that depend on U34 modification do not predict protein expression. They then demonstrate that a specific gene codon-content is required but not sufficient to predict protein aggregation. In addition, the protein should contain a hydrophilic penta sequence motif for aggregation to occur. They also show that triple negative breast cancers require U34 enzymes for cell survival and determine the proteome subset most affected by U34 tRNA modifying enzymes.

The manuscript is interesting, however, there are some concerns that should be addressed.

Major concerns

- The mechanism by which the described motif promotes protein aggregation should be further explored and discussed. It has been described that flanking sequences of aggregation-promoting regions are enriched in amino acids with low aggregation properties that will oppose aggregation and favor chaperone binding (Rousseau F, Serrano L, Schymkowitz JW ,2006. J Mol Biol 355: 1037–1047). These amino acids include precisely R, K, D and E, the four amino acids present in the identified motif. Is thus the motif just a signal for the presence of aggregation promoting regions? In such case, it would make sense that a decrease of translation elongation in U34 codon-rich mRNAs triggered by ELP3 depletion would specially affect proteins with a tendency to aggregate.
- Translation elongation speed has been linked to amino acid composition of the synthesized protein. Can one eliminate the possibility that hydrophilic regions might have an effect on translation elongation and/or folding by interacting with the ribosome channel or surface?
- The addition of the data related to the triple negative breast cancer does not seem to add value to the manuscript. It was already established that genes related to cell cycle regulation are enriched in U34 codons and that abnormal expression of U34 tRNA modifying enzymes are found and linked to different tumors. Moreover, it does not add anything related to the mechanism linked to the observed protein aggregation.
- Total expression of KIF4A or KIF14 does not seem to change when considering together the signals from WCE and aggregates (fig 1e or 4d). How was the sample of the proteomic data (Fig 1 d) obtained? Were the aggregates lost because of the method of extraction used?

Minor concerns

- Extended Fig 1. Introduce it briefly in the text.
- Line 119, change transcription levels by mRNA steady-state levels
- Line 124 Fig 1e-f does not show KIF23 expression. Likewise, KIF23 mRNA levels are not shown in 1g or extended figures.
- Fig 2f is not mentioned in the text. In the figure legend, the meaning of the used colors should be clarified.
- Related to Fig. 3. Move Fig 3a, b and c to extended figure 3. Move Fig. 5b from extended figures to main figures. Eliminate Fig 3d (indicate the data directly in the text).
- The manuscript is too compressed. It will benefit from a more extended version.

Reviewer #2 (Remarks to the Author):

The focus of the article is to determine if enrichment in XAA codons is sufficient to determine which proteins are the most affected by suppression of mcm5s2U34 synthesis. The authors initially validate that 5 human proteins encoded by the most XAA enriched mRNAs are indeed significantly downregulated following ELP3 and/or CTU depletion. They then proceeded to perform mass spectrometry and transcriptome analysis in control and ELP3-depleted cells. This analysis revealed that XAA enrichment poorly correlates with changes in protein abundance following ELP3 depletion. At this point, somehow inexplicably, the focus seems to change from transcriptome-wide and proteomic-wide analysis to kinesins. This progression from all transcripts to kinesins specifically is unclear. There is no clear justification that kinesins were selected as having the biggest changes in protein abundance by mass spectrometry (i.e. fold changes for these kinesins in Supp Table 2 are relatively low and do not seem to correlate with western blot results in Fig 1F showing significant depletion). This raises the possibility that the mass spectrometry coverage was not good enough to detect with confidence the majority of proteins affected by ELP3 depletion.

In spite of these limitations in the cohesion of the data presented, the article is interesting because it identifies for the first time a specific cluster of amino acids responsible for lower abundance of a subset of proteins encoded by "U34-rich transcripts" when U34 modifying enzymes are depleted. The last section focussed on TNBC seems disconnected from the main body of the article. The authors have previously shown the importance of U34-modifying enzymes in breast cancer models and in melanoma models recently (Rapino, Nature 2018). The TNBC results are not linked to the newly identified pentamer or to aggregation of proteins being a reason for increased sensitivity to U34 pathway inhibition compared to ER-positive disease.

Other concerns:

The method to identify XAA enriched transcripts is odd. I do not see a biological reason why transcript such as HMG5 with a 35% content in XAA codons (18% GAA, 15% AAA and 1% CAA) would be excluded from the "top transcripts", same for SREK1P1 with 30%. Both mRNAs have higher enrichment than MNS1. The data should be replotted to show cumulative relative content of all three codons in all human transcripts.

Same concept also applies for transcripts without any XAA codons, why are ELFN2 and CELSR3 selected as they actually have minimal XAA codons? There are at least 100 human transcripts with no XAA codons at all, including PEX10 and DRD4.

Another curious oversight is that it not even mentioned if the proteins in Figure 1C contain the hydrophilic pentasequence. This leaves the reader wondering if there are proteins without hydrophilic pentamers which are highly affected by ELP3 and/or CTU2 depletion. It should be made clear if MNS1, NEXN, HMMR, CEP290 and CEP83 contain the pentamer.

Line 114: Authors should avoid saying "in the absence of ELP3", this is not a knockout, it is a knockdown.

It is not clear what the dynamic range to detect changes in protein abundance is in the proteomic data. What is the fold change for ELP3 in the MS dataset? ELP3 protein should be indicated clearly on the MS plot. The axis for proteomics covers -0.3 to 0.3 so a maximum change of 1.23 fold each way. How does the mass spec fold change compare with the western blot fold change for KIF15, KIF4A and KIF14? Were proteins from Fig1c detected by mass spectrometry?

It is not possible to exclude that changes in protein stability following ELP3 knockdown may explain the absence of changes in KIF5B or KIF23 abundance. Half-life experiments should be performed on at least 1 or 2 U34-enriched candidates.

Reviewer #3 (Remarks to the Author):

In this manuscript, Rapino et al. explained the potential regulatory mechanisms that interconnect

translational regulation of the mRNAs and the protein fate/expression levels. To this end, the authors utilized 'loss-of-function models of U34-enzymes' and measured the changes in protein expression level via proteomics and biochemical approaches. To gain a global view on the impact of U34 modification on protein translation, the authors analyzed the total cellular proteome with or without ELP3 knockdown, and monitored changes in protein expression. From the data, authors emphasize that proteins downregulated upon depletion of ELP3 are enriched in U34 codon-bearing genes (while transcript levels remain largely unaltered), although not all U34-codon bearing genes are downregulated (Fig. 1d-e). Importantly, the authors find that kinesin proteins (U34 codon-bearing) are downregulated in protein but not mRNA level, which leads to further dissection of individual kinesin proteins in the following experiments.

Regarding the technical validity of proteomics data, the data interpretation seems to have three major misleading points:

- 1) The fold changes in protein expression seem too small (within 0.1 in log₂ scale, according to the Fig. 1d-e and Ext Fig. 2g) to discuss any significant changes in expression level ($< 2^{0.1} = \sim 1.08$; in fact, the RNA-seq expression levels show much bigger changes of $< 2^3 = 8$). This level of fold change should be interpreted as 'No-change', considering the general quantification accuracy of MS intensity based LFQ approach.
- 2) The classification of 'Up-regulated', 'No-change', and 'Downregulated' seem a bit skewed (No-change should be of equal width in both directions).
- 3) The western blot results in Fig. 1f shows considerable changes of protein levels for KIF4A/14/15, but would not agree to the proteomics data (if they were identified and quantified; the authors should mark the proteins within the scatter plot).

Regarding the issues raised above, the authors should refrain from making any conclusions regarding protein expression level changes from the "current" proteomics data.

Nonetheless, Fig 1f alone seems sufficient to continue dissection into the roles of U34 codons in translation of kinesin proteins. In this case, however, the authors should take a step back from arguing any global effects of U34 modifications, but put emphasis on the regulation of specific subsets of proteins that have U34 codons (In fact, the author's initial description that some but not all U34-codon bearing proteins are downregulated is itself confusing to discuss the global roles of ELP3).

Rebuttal letter

We thank referees for the constructive evaluation of our work. As requested, we have now extensively revised the manuscript according to the referee's comments. The manuscript now encloses 8 main figures, complemented with 8 supplementary figures and 4 supplementary tables. The figures were redesigned to 1/ include the new data, 2/ remove the TNBC data, and 3/ include the data shown in former supplementary figure 1, as requested. Please see our detailed answers here below:

Reviewer #1 (Remarks to the Author):

It is established that changes in translation elongation speed caused by depletion of U₃₄ tRNA modifying enzymes elicit a widespread protein aggregation in vivo. Rapino et al. here take a step ahead. First, they observe by combining RNA-seq and proteome analyses that an enrichment in codons that depend on U₃₄ modification does not predict protein expression. They then demonstrate that a specific gene codon-content is required but not sufficient to predict protein aggregation. In addition, the protein should contain a hydrophilic pentasequence motif for aggregation to occur. They also show that triple negative breast cancers require U₃₄ enzymes for cell survival and determine the proteome subset most affected by U₃₄ tRNA modifying enzymes.

The manuscript is interesting, however, there are some concerns that should be addressed.

We thank the reviewer for her/his positive assessment of our manuscript and for her/his comments.

Major concerns

- The mechanism by which the described motif promotes protein aggregation should be further explored and discussed. It has been described that flanking sequences of aggregation-promoting regions are enriched in amino acids with low aggregation properties that will oppose aggregation and favor chaperone binding (Rousseau F, Serrano L, Schymkowitz JW ,2006. J Mol Biol 355: 1037–1047). These amino acids include precisely R, K, D and E, the four amino acids present in the identified motif. Is thus the motif just a signal for the presence of aggregation promoting regions? In such case, it would make sense that a decrease of translation elongation in U₃₄ codon-rich mRNAs triggered by ELP3 depletion would specially affect proteins with a tendency to aggregate.

We thank the reviewer for her/his suggestion. Please find below a description of the work we performed to address this question. We added new panels in the manuscript (**new Fig. 7 and new Suppl Fig. 8**) and we also

discuss the different hypotheses on the mechanism by which the identified motif promotes protein aggregation upon U₃₄-enzymes depletion in the new version of the manuscript (lines 334-368).

First, to assess if the identified motif is a **signal for the presence of aggregation promoting regions**, as proposed by the reviewer, we performed a series of analyses:

In the original version of the paper, we calculated protein aggregation propensity by using the *Zygggregator* method (Z_{agg}). We showed that U₃₄-target proteins (i.e.: proteins whose mRNA is enriched in U₃₄-codons and that are downregulated upon U₃₄-enzymes depletion) are not intrinsically more prone to aggregation as compared to proteins that possess similar frequency of U₃₄-codons but whose expression levels remain unchanged upon U₃₄-enzymes depletion (Suppl Fig. 5c). We further confirmed this observation by assessing each of the parameters therein individually. Unexpectedly, the hydrophobic score (Suppl Fig. 5e) and the presence of patterns of five hydrophobic amino acids (Suppl Fig. 5f) were very similar in protein targets and non-targets of U₃₄-enzymes. Only the presence of patterns of five hydrophilic amino acids was able to discriminate the two protein populations. To more specifically answer the question of the reviewer, we now systematically assessed the association between hydrophilic pentamers and 1/ aggregation propensity (REV#1 Fig. 1a); 2/ hydrophobic score (REV#1 Fig. 1b); and 3/ hydrophobic pentamers (REV#1 Fig. 1c) in target and no target proteins by calculating spearman correlations. These analyses highlighted that there is no significant correlation between the presence of hydrophilic pentamers and any of the known aggregation prone characteristics in target proteins ($R^2= 0.1$, $p=0.02$; $R^2=0.009$, $p=0.09$; and $R^2= 0.016$, $p=0.33$, respectively). Previous studies highlighted the potential role of hydrophilic stretches as gatekeepers for the correct folding of nascent proteins due to their proximity to aggregation prone regions (*Pawar et al, J.Mol.Biol. 2005, 350(2):379-92*; *Rousseau F et al, J. Mol Biol 2006, 355(5):1037-47*). Therefore, we assessed if the identified hydrophilic pentamer is in proximity to aggregation prone regions in KIF4A by using two independent predictive algorithms: Z_{agg} and TANGO (*Fernandez-Escamilla et al, Nat. Biotechnol. 2004, 22:1302-1306*). We found that the identified hydrophilic pentamer is in proximity to **low** aggregation-prone regions (REV#1 Fig1. d-e). Moreover, the deletion of the motif in KIF4A-Del, which inhibits KIF4A aggregation upon U₃₄-enzymes depletion, does not affect the global aggregation propensity of KIF4A as assessed by the TANGO algorithm (REV#1 Fig. 1f). Together, although they do not completely exclude this possibility, our analyses failed to support the idea that the function of the identified motif is a signal for the presence of aggregation promoting regions. These data are not shown in the manuscript for sake of clarity, but they are discussed in lines 340-344.

REV#1 Figure 1 legend:

a, b, c, Dot plot of aggregation propensity (a), hydrophobic score (b) or hydrophobic pentamers (c) and number of hydrophilic pentamers in proteins enriched in U₃₄-codons ($q < 0.05$) downregulated or not upon U₃₄-enzymes depletion (Spearman correlation was calculated). **d,** KIF4A aggregation propensity was calculated per each amino acid using the zyggregator method (Zagg). Orange bar identify the hydrophilic motif. **e,** Beta aggregation propensity was calculated for each amino acid of KIF4A by TANGO algorithm. **f,** Global propensity aggregation (beta-aggregation) of KIF4A-WT and KIF4A-Del assessed by the TANGO algorithm for each bin (= 60 aa); two-sided *t*-test.

To further understand the mechanisms by which the described motif may promote protein aggregation, we performed a series of additional analyses to address if the motif 1/ is a **binding sequence** for proteins that promote/resolve aggregation; 2/ is a signal that promotes or cooperates with **chaperone-mediated autophagy** (CMA); 3/ mediates aggregation through **its charged/ampholytic nature**. We developed each of the hypotheses here below:

1/ Our analyses exclude that the identified hydrophilic motif is a binding sequence for proteins that promote/resolve aggregation (REV#1 Fig. 2a-c).

The motif strongly differs from known chaperone- and cochaperone- binding sites, which are generally hydrophobic (*Koldewey et al, Cell 2016, 166(2):369-379*). Nevertheless, we calculated the protein binding propensity along the KIF4A sequence using SCRIBER (*Zhang and Kurgan, Bioinformatics 2019; <http://biomine.cs.vcu.edu/servers/SCRIBER/>*) (REV#1 Fig. 2a). The hydrophilic motif (in orange) is enclosed in a region presenting one of the lowest propensity to bind proteins, as compared to the average binding propensity along the KIF4A sequence. It is therefore unlikely that the motif serves as binding region to aggregation-mediator proteins. Next, we assessed if the identified motif in KIF4A could serve as a chaperone-binding region by using the ChaperISM-predictive algorithm (*Gutierrez et al, Bioinformatics 2020, 36(3):735-741*). As expected, we found many HSP70-putative binding sites across the KIF4A sequence, including the at the location of identified hydrophilic motif (REV#1 Fig. 2b). To understand the significance of this predicted overlap, we questioned 1/ if the deletion of the motif in KIF4A (i.e. in KIF4A-Del) significantly affected the general binding propensity of HSP70 on the protein and 2/ if we could highlight an increased propensity to bind HSP70 in proteins downregulated upon ELP3-depletion. To answer these questions, we first evaluated the HSP70 binding propensity in KIF4A-WT and KIF4A-Del by ChaperISM-algorithm. Our analysis failed to highlight any significant change in HSP70-binding propensity between the two proteins (ttest, $p=0.673$; data not shown). Second, we performed GSEA analyses with our proteomic dataset and we could not highlight any significant enrichment in HSP70 binding signature in proteins downregulated upon ELP3-depletion (REV#1 Fig. 2c). These analyses failed to provide any supporting evidence that the motif is a binding site for proteins that promote/resolve aggregation upon the depletion of U₃₄-enzymes. These data are not shown in the manuscript for sake of clarity.

REV#1 Figure 2 legend:

a, KIF4A protein binding propensity was calculated by SCRIBER. Orange line identifies the hydrophilic pentamer. **b**, HSP70 putative binding sites were predicted on KIF4A sequence by ChaperISM (quantitative mode), blue dots identify significant predicted binding aminoacids, red dots represent the hydrophilic motif. **c**, GSEA analysis of proteomics of the BT549 cells depleted or not of ELP3. **d**, Chaperone mediated autophagy (CMA) motifs distribution in proteins enriched in U₃₄-codons ($q < 0.05$) downregulated or not upon U₃₄-enzymes depletion. Two-side ttest. **e**, **f**, Dot plot of CMA motifs and number of hydrophilic pentamers in proteins enriched in U₃₄-codons ($q < 0.05$) downregulated (**f**) or not (**e**) upon U₃₄-enzymes depletion (Spearman correlation was calculated).

2/ We also exclude that the identified motif may promote or cooperate with chaperone-mediated autophagy (CMA).

The motif we identified in this study shares some similarities with the motifs responsible for CMA (i.e. KFERQ and derivatives; *Kirchner et al, PLoS Biol. 2019, 17(5):e30000301*). Also, we previously showed that target proteins of U₃₄-enzymes are downregulated mainly because they are degraded through a lysosome-dependent pathway (*Rapino et al, Nature 2018, 558(7711):605-609* and Suppl Fig. 3a). Still, an important difference is seen between our observations and the CMA hypothesis: the substrates recognized and degraded by CMA are not described to undergo aggregation before integration into lysosomes (*Kaushik and Cuervo, Nat.Rev.Mol.Cell Biol. 2018, 19(6):365-381*). In our study, we link defective codon-dependent translation to protein aggregation and subsequent degradation. Moreover, we showed that differences in protein output upon U₃₄-enzymes depletion is mainly explained by differences in protein aggregation, placing this process at the center of our mechanism. Nevertheless, we quantified the putative presence of CMA motifs in target and not target proteins of U₃₄-enzymes by using the KFERQ-finder software (*Kirchner et al, PLoS Biol. 2019, 17(5):e30000301*) (REV#1 Fig. 2d-e). These analyses failed to highlight any difference in the presence of CMA motifs between the two groups of proteins, nor any significant association between CMA motifs and the hydrophilic motif described in this study. Therefore, we concluded that it is unlikely that chaperone-mediated autophagy is involved in the degradation of protein targets upon depletion of U₃₄-enzymes. These data are shown in the manuscript in Suppl Fig. 8d-e and briefly discussed in the revised manuscript (lines 336-340).

3/ We hypothesized that the charged/ampholytic nature of the identified hydrophilic motif mediates protein aggregation upon depletion of U₃₄-enzymes.

The identified hydrophilic pentamer motif is defined by its unique amino acid composition, exclusively enclosing charged residues, but also representing ampholytic motifs. We surmised that this unique feature might be the reason why proteins are subjected to aggregation upon depletion of U₃₄-enzymes.

Recent evidences described the propensity of low complexity regions (LCR) rich in charged amino acids to regulate protein phase behavior, to promote liquid-liquid phase separation (LLPS) and to trigger protein aggregation (*Xue et al, Sci.Adv. 2019, 5(11):eaax5349*; *Lin et al, Biochemistry 2018, 57(17):2499-2508*; *Franzmann et al, J.Biol.Chem. 2019, 294(18):7128-7136*; *Babinchak and Surewicz, J.Mol.Biol. 2020, 432(7):1910-1925*; *Molliex et al, Cell 2015, 163(1):123-33*). LLPS is a process in which proteins or protein-RNA complexes spontaneously segregate in response to stress. P-bodies and stress granules are examples of well-described structures formed through LLPS (*Babinchak and Surewicz, J.Mol.Biol. 2020, 432(7):1910-1925*). In the Xue et al paper (*Xue et al, Sci.Adv. 2019, 5(11):eaax5349*), the authors highlight the capability of two LCRs in the U1-70K protein to drive LLPS and aggregation in cells. Furthermore, they highlight the importance of the presence of positively and negatively charged amino acids to promote this process. Strikingly, the motif

identified in our study shares similar physical characteristics with the U1-70K LCRs domains. **Therefore, we postulated that the presence of the hydrophilic motif might trigger similar LLPS and auto-aggregation among the U₃₄-target proteins, upon U₃₄-enzymes depletion.** To support this hypothesis, we further analyzed the proteins found in aggregates upon ELP3 depletion (see Fig. 3d). Interestingly, we found a significant enrichment in proteins harboring a RRM1 (RNA recognition motif 1) domain in the aggregates induced by ELP3 depletion (REV#1 Fig. 3a), suggesting some similarities with the LLPS bodies (*Banani et al, Cell 2016, 166(3):651-663*). Moreover, we visualized KIF4A-WT or KIF4A-Del (i.e. that does not harbor the hydrophilic motif) proteins in MCF7 cells by immunofluorescence. Interestingly, upon depletion of U₃₄-enzymes, KIF4A-WT aggregates in small punctuated structures, similar to SHP2 mutants-LLPS associated structures (*Zhu et al, Cell 2020, 183(2):490-502*). Moreover, KIF4A-Del is never found in these structures in U₃₄-enzymes depleted cells (**new Fig. 7c**; REV#1 Fig. 3b). Finally, we assessed the amyloid aggregation propensity in KIF4A-WT and KIF4A-Del using the TANGO algorithm (*Fernandez-Escamilla et al, Nat. Biotechnol. 2004, 22(10):1302-6*) (REV#1 Fig. 3c). Interestingly, the hydrophilic motif we identified in KIF4A is predicted to have high propensity to undergo amyloid aggregation. Moreover, its deletion (i.e. KIF4A-Del) causes a net reduction in amyloid aggregation propensity in KIF4A, as measured by the total area under the curve (KIF4A WT= 163231 and KIF4A Del= 146070), and a change in the maximal amyloid aggregation peak (KIF4A-WT max peak – intensity 34359- is found in bin 9 that harbors the motif, while KIF4A-Del max peak – intensity 24352- is found in bin 14).

These observations indicate that the presence of the charged/ampholytic hydrophilic motif may trigger aggregation through changes in protein phase behavior and LLPS upon depletion of U₃₄-enzymes. This suggests that the translational defects observed at all the tested U₃₄-enriched transcripts upon depletion of U₃₄-enzymes might result in protein downregulation, only if the exposure of charged/ampholytic motifs is sufficient to trigger LLPS driven aggregation that will lead to clearance by the lysosome compartment. Further experiments will need to be done to clarify if these hypotheses are correct.

These data are now added in the manuscript (**new Fig. 7c-d**) and the hypothesis is further developed in the discussion part of the revised manuscript (344-356).

REV#1 Figure 3 legend:

a, GSEA analysis of proteomics of the aggregates extracted from BT549 cells depleted or not of ELP3. **b**, Immunofluorescence of MCF7 cells depleted or not of ELP3 and overexpressing KIF4A-WT or KIF4A-Del in control or after 6h of treatment with 100 μ M chloroquine. Arrows depict KIF4A aggregates. **c**, Amyloid aggregation profiles of KIF4A-WT and KIF4A-Del calculated by using the TANGO algorithm.

- Translation elongation speed has been linked to amino acid composition of the synthesized protein. Can one eliminate the possibility that hydrophilic regions might have an effect on translation elongation and/or folding by interacting with the ribosome channel or surface?

As the reviewer pointed out, interactions of the nascent peptide with the ribosome channel influence ribosome speed. The ribosome exit tunnel is a negatively charged cavity from which the nascent proteins arise. Different factors are involved in facilitating or opposing the exit of the nascent peptide from the ribosome tunnel, among these tRNA availability, protein secondary structure formation and the presence of charged amino acids in the nascent peptide (Sharma et al, *Phys. Rev. E* 97, 022409; Tuller et al, *Genome Bio.* 2011, 12(11):R110;; Gorochoowski et al, *Nucleic Acids Res.* 2015, 43(6):3022-3032). Positively charged amino acids (K, R, H) can interact with the ribosome tunnel surface altering the profiles of the forces acting on the nascent chain and the local energetic potential barriers that needs to be overcome to effectively exit nascent peptides from the ribosome exit tunnel. The identified hydrophilic pentamers are rich in positively charged amino acids. Consequently, we found that U₃₄-enriched proteins downregulated upon ELP3 depletion, display a significantly higher presence of K, H, R islands, as compared to U₃₄-proteins whose expression remains unchanged in shELP3 cells (REV#1 Fig. 4a). We recently computed the ribosome exit tunnel electrostatics in a model to quantify the energetic needs required to effectively exit nascent peptides from the ribosome tunnel (Joiret et al, *bioRxiv* 2020.10.20.346684; doi: <https://doi.org/10.1101/2020.10.20.346684>; submitted). Using this model, we show that the presence of the identified hydrophilic motif in the KIF4A protein sequence represents a region of high energetic needs in order to exit the ribosome tunnel and to progress towards translation elongation (**new Fig.7a**; Editor Fig. 3a). Conversely, the deletion of the hydrophilic motif in the KIF4A protein sequence (KIF4-Del; see Fig. 4a), which prevents KIF4A aggregation upon depletion of U₃₄-enzymes, alters the local energetic barrier, which is predicted to favor ribosome movement across this region (**new Fig.7b**; Editor Fig. 3b). Upon depletion of U₃₄-enzymes, transcripts enriched for U₃₄-codons show accumulation of ribosomes as a result of ribosome pausing (Nedialkova & Leidel, *Cell* 2015, 161(7):1608-1618; Rapino et al, *Nature* 2018; 558(7711):605-609; and Fig. 1d, 2d, 3h). It is therefore tempting to speculate that the additional energetic barrier brought by the presence of the hydrophilic motif could be the cause of protein misfolding and aggregation. **We now added these data (new Fig. 7a-b) and we discussed this hypothesis in the revised version of the manuscript (lines 357-368).**

REV#1 Figure 4 legend:

a, Empirical cumulative distribution function of the number of RKH islands encountered in U_{34} -codons enriched proteins classified by protein expression levels upon ELP3 depletion. Kolmogorov-Smirnov test rejecting the null (p -value=0.04) that the number of RKH islands are the same in the 2 groups. The number of RKH islands is significantly higher in the proteins for which the expression levels are lower upon ELP3 depletion. **b**, **c**, Axial forces profile on nascent chain caused by the electrostatic interaction in the ribosome exit tunnel b) wild type KIF4A; c) ERRRK deleted KIF4A. The axial forces profiles and local mechanical work acting on the two chains are different upon incorporation of residues 596-646 at the peptidyl-transferase center.

- The addition of the data related to the triple negative breast cancer does not seem to add value to the manuscript. It was already established that genes related to cell cycle regulation are enriched in U34 codons and that abnormal expression of U34 tRNA modifying enzymes are found and linked to different tumors. Moreover, it does not add anything related to the mechanism linked to the observed protein aggregation.

We followed the suggestion of the reviewer. We removed the TNBC data in the revised manuscript.

- Total expression of KIF4A or KIF14 does not seem to change when considering together the signals from WCE and aggregates (fig 1e or 4d). How was the sample of the proteomic data (Fig 1 d) obtained? Were the aggregates lost because of the method of extraction used?

The blots detecting KIF4A and KIF14 in WCE and aggregate fractions (see new Fig. 3e) should not be interpreted as if the two compartments are shown in a stoichiometric manner. Indeed, 30µg of protein were loaded in WCE while the aggregates extracted from 1mg WCE were loaded in the aggregate fraction, therefore having 40 times more concentrated sample. We clarified this aspect in the legend of the new Fig. 3e. To specific answer the question related to extraction, cells were lysed in a SDS 2 % buffer (50 mM Tris-HCl pH 8, 2% SDS, 150 mM sodium chloride, 10 mM sodium fluoride and 1 mM trisodium phosphate).

Minor concerns

-Extended Fig 1. Introduce it briefly in the text.

We updated the revised manuscript with a description of the data in the text (lines 100-107). Moreover, we now included two panels of the former supplementary figure 1 in the main figure 1 (new Fig. 1a-b). We hope that this new version is more clear.

-Line 119, change transcription levels by mRNA steady-state levels

We changed the test as suggested (line 137 of revised manuscript)

- Line 124 Fig 1e-f does not show KIF23 expression. Likewise, KIF23 mRNA levels are not shown in 1g or extended figures.

We detected KIF23 protein and RNA levels as suggested from the reviewer (REV#1 Fig.5a-g). We now included the data in the revised manuscript (Fig. 2a, 2c, 2d and Suppl Fig. 2a, 2b, 2d, 2e,2f).

REV#1 Figure 5 legend:

a, b, Volcano plot of the proteomics (n= 3 independent experiments, a) and RNA-seq (n= 3 independent experiments, b) of BT549 cells depleted or not of ELP3. KIF23 is highlighted. **c, d, e**, KIF23 protein (c) and mRNA levels (d, e) were assessed in MDA-MB231 and BT549 by western blot or RT-PCR, respectively.

-Fig 2f is not mentioned in the text. In the figure legend, the meaning of the used colors should be clarified.

We now mentioned Fig. 2f in the revised manuscript (now Fig. 3f, line 186). We also included the description of the color code used in the figure legend of the revised manuscript. Specifically, the different colors identify the functional domains of KIF4A protein: in green the Kinesin Motor Domain, in grey the Coiled-Coil and in blue the Globular Region.

- Related to Fig. 3. Move Fig 3a, b and c to extended figure 3. Move Fig. 5b from extended figures to main figures. Eliminate Fig 3d (indicate the data directly in the text).

We made the suggested changes and we updated the revised manuscript accordingly.

- The manuscript is too compressed. It will benefit from a more extended version.

We extensively revised the manuscript. The manuscript is now composed of 8 main figures, 8 supplementary figures and 4 supplementary tables. We removed the data related to TNBC, and we extended the results and the discussion sections, integrating the points highlighted by the reviewer and discussing more extensively the possible roles of the identified motif. We hope that the reviewer will find this version more clear.

Reviewer #2 (Remarks to the Author):

The focus of the article is to determine if enrichment in XAA codons is sufficient to determine which proteins are the most affected by suppression of mcm5s2U34 synthesis. The authors initially validate that 5 human proteins encoded by the most XAA enriched mRNAs are indeed significantly downregulated following ELP3 and/or CTU depletion. They then proceeded to perform mass spectrometry and transcriptome analysis in control and ELP3-depleted cells. This analysis revealed that XAA enrichment poorly correlates with changes in protein abundance following ELP3 depletion.

At this point, somehow inexplicably, the focus seems to change from transcriptome-wide and proteomic-wide analysis to kinesins. This progression from all transcripts to kinesins specifically is unclear. There is no clear justification that kinesins were selected as having the biggest changes in protein abundance by mass spectrometry (i.e fold changes for these kinesins in Supp Table 2 are relatively low and do not seem to correlate with western blot results in Fig 1F showing significant depletion). This raises the possibility that the mass spectrometry coverage was not good enough to detect with confidence the majority of proteins affected by ELP3 depletion.

We thank the reviewer for her/his comments. We heard the reviewer concern and we now discuss more extensively the choice to focus on the kinesin family (lines 122-126 and lines 147-148; Suppl Fig. 1j). As the reviewer pointed out, the fold changes in the proteomic were calculated from the average LFQ intensity of three independent experiments. Log₂ of the delta of log₂ LFQ shELP3 and log₂ LQF control cells (shELP3/shCTR) was then calculated, resulting in a very small fold change range. We apologize for this mistake. We now replotted the data by calculating the difference of the average of LFQ intensity (log₂) between shELP3 cells and shCTR (see REV_Table#1, calculation done with Prism graph pad v.8). The new scale is consequently wider and better reflects the changes we observed by western blot. The new Supplementary Table 1 has been corrected accordingly.

We decided to focus on the kinesin family for the following reasons: first, we performed a GSEA analysis, which highlighted that Kinesins were significantly downregulated upon ELP3 depletion in our proteomic dataset (Rev#2 Fig. 1a). Second, the kinesin family shows a significant enrichment in proteins whose RNA is strongly enriched in U₃₄-codons compared to human genome (Rev#2 Fig. 1b). Third, among the kinesins whose mRNA is enriched of U₃₄-codons, we could highlight difference in behavior upon depletion of ELP3 in our proteomic. As expected, some kinesins enriched in U₃₄-codons such as KIF4A, KIF14 or KIF15 were downregulated upon ELP3 depletion. Nevertheless, we found kinesins that display similar frequency in U₃₄-codons (i.e. KIF5B or KIF23) but whose expression remained unchanged in the same dataset).

REV#2 Figure 1 legend:

a, GSEA analysis of proteomics of the BT549 cells depleted or not of ELP3. **b**, AAA&GAA&CAA cumulative frequency of the members of the kinesin family, red dots depict enriched members (frequency > 0.088). U₃₄-codon rich transcripts enrichment was calculated by χ^2 test compared to the genome.

In spite of these limitations in the cohesion of the data presented, the article is interesting because it identifies for the first time a specific cluster of amino acids responsible for lower abundance of a subset of proteins encoded by “U34-rich transcripts” when U34 modifying enzymes are depleted.

We thank the reviewer for her/his positive comment.

The last section focused on TNBC seems disconnected from the main body of the article. The authors have previously shown the importance of U34-modifying enzymes in breast cancer models and in melanoma models recently (Rapino, Nature 2018). The TNBC results are not linked to the newly identified pentamer or to aggregation of proteins being a reason for increased sensitivity to U34 pathway inhibition compared to ER-positive disease.

We followed the reviewer recommendation. We removed the data related to TNBC.

Other concerns:

-The method to identify XAA enriched transcripts is odd. I do not see a biological reason why transcript such as HMG5 with a 35% content in XAA codons (18% GAA, 15% AAA and 1% CAA) would be excluded from the “top transcripts”, same for SREK1P1 with 30%. Both mRNAs have higher enrichment than MNS1. The data should be replotted to show cumulative relative content of all three codons in all human transcripts. Same concept also applies for transcripts without any XAA codons, why are ELFN2 and CELSR3 selected as they actually have minimal XAA codons? There are at least 100 human transcripts with no XAA codons at all, including PEX10 and DRD4.

We understand the reviewer concern. The quantification of codon content in mRNA can be done by calculating 1/ the cumulative codon frequency per gene (REV#2 Fig. 2a) or 2/ the codon usage (REV#2 Fig.2b). While the cumulative codon frequency highlights the number of codons related to mRNA length used in a specific gene, the codon usage analysis assesses the times a specific amino acid is coded by one of its synonymous codons. Therefore, the use of a cumulative frequency method leads to an arbitrary choice of the enrichment threshold (i.e.: lower limit of the 3rd quartile, REV#2 Fig. 2a), while the use of a codon usage method sets enrichment at ratios >1 (REV#2 Fig. 2b), neglecting the number of codons present in the mRNA. The method here developed aims to quantitatively determine the mRNA codon enrichment score from a frequency perspective **by the assessment of a q value** (REV#2 Fig. 2c). Briefly, we calculated the frequency of a specific codon (i.e. here AAA, GAA and CAA) used in a gene and compared it with the codon frequency in a gene randomly picked in the human genome, to assign statistical significance. 10.000 times comparison were made for each gene of the human genome. A p_{more} value (i.e. the possibility that a gene of interest has higher frequency of the specific codons than a randomly picked gene) was calculated and used to generate a **q-value** for each gene. A cut off of q=0.05 was used for each U₃₄-codon to determine significant bias in codon content (REV#2 Fig. 2c; shown as new figure 1c). Genes such as HMGN5 (q-value AAA=0; GAA=0, CAA=1) are therefore excluded as enriched genes because they do not fulfil the q-value<0.05 for all three codons. We then compared the different methods (REV#2 Fig.2c-d). We could highlight that the genes classified as enriched with our method are all in the 3rd quartile of the cumulative codon frequency distribution of the human genome. Therefore, as the reviewer remarked, the exclusion of some genes harbouring high frequency of the U₃₄-codons is not based on biological reasons but on the author choice to increase the cut off stringency. To strengthen our conclusions that codon content is not sufficient to predict protein fate, we have performed the some key analyses using the simple cumulative frequency (of U₃₄-codons). These analyses are included in the new version of the MS (Fig. 1c, 2a, 2b, 3d and 4d) and support the validity of our conclusions independently on the method of enrichment used.

REV#2 Figure 2 legend:

a, b, c, Humane genome wide analysis of U₃₄-codons enrichment by cumulative frequency (cut off 3rd quartile= 0.088, a); codon usage (cut off= 1, b) or q value (cut off q<0.05). **d,** Comparative table of the three U₃₄-codons enrichment methods.

- Another curious oversight is that it not even mentioned if the proteins in Figure 1C contain the hydrophilic pentasequence. This leaves the reader wondering if there are proteins without hydrophilic pentamers which are highly affected by ELP3 and/or CTU2 depletion. It should be made clear if MNS1, NEXN, HMMR, CEP290 and CEP83 contain the pentamer.

We thank the reviewer for this question. We now added the missing information in the new version of the manuscript (Line 330-333, **new Supplementary Figure Fig. 8c**). Strikingly, hydrophilic pentamers were found in all the proteins of former Figure 1c (REV#2 Fig. 3a): MNS1= 9 pentamers, NEXN= 15 pentamers, HMMR= 5 pentamers, CEP83= 5 pentamers and CEP290=14 pentamers. Importantly, hydrophilic pentamers were also found in all the previously identified targets of the U₃₄-enzymes: SOX9 (*Ladang et al, J.Exp.Med. 2015, 212(12):2057-75*), DEK (*Delaunay et al, J.Exp.Med. 2016, 213(11):2503-2523*), HIF1 α (*Rapino et al, Nature 2018, 558(7711):605-609*) and hnRNPQ (*Xu et al, Nat. Comm. 2019, 10(1):5492*) (REV#2 Fig. 3a).

As the reviewer pointed out, there are some U₃₄-target proteins (i.e.: enriched in U₃₄-codons and downregulated upon depletion of U₃₄-enzymes) that do not harbor any hydrophilic pentamer (REV#2 Fig. 3b;

88% with pentamer, 12% without pentamer). One example assessed in this manuscript, is the kinetochore protein NUF2. By using a codon-mutant construct of NUF2, where all the U₃₄-sensitive codons are replaced by their synonymous counterpart (new Fig. 6a), we demonstrate that NUF2 is not a translational target of the U₃₄-enzymes. We therefore speculate that NUF2 downregulation might be the consequence of the U₃₄-mediated downregulation of NDC80, its kinetochore binding partner. NDC80 has been described to bind NUF2 and they mutually regulate their stability. As NUF2, NDC80 is enriched in U₃₄-codons and downregulated upon ELP3 depletion, but it also harbors hydrophilic pentamers (new Fig. 6a). Importantly, the use of a NDC80 codon mutant rescued NDC80 expression upon depletion of U₃₄-enzymes (new Fig. 6b), identifying NDC80 as the direct target of U₃₄-enzymes. Importantly, the normal expression of NDC80 (i.e. codon mutant) in ELP3-depleted was sufficient to rescue NUF2 protein levels, indicating that NUF2 expression relied on that of NDC80 and that NUF2 is not a direct target of U₃₄-enzymes (new Fig. 6e). This set of data demonstrates that the presence of hydrophilic pentamers is a novel feature to discriminate the mcm⁵s²U₃₄-dependent proteome.

REV#2 Figure 3 legend:

a, Schematic representation of proteins whose expression depends on the U₃₄-enzymes. Pink boxes represent hydrophilic pentamers. **b**, Schematic representation of the percentage of U₃₄-enriched proteins (q<0.05) downregulated upon ELP3 depletion harboring (blue) or not (grey) at least one hydrophilic pentamer.

Line 114: Authors should avoid saying “in the absence of ELP3”, this is not a knockout, it is a knockdown.

We changed this across the manuscript as requested.

- It is not clear what the dynamic range to detect changes in protein abundance is in the proteomic data.

What is the fold change for ELP3 in the MS dataset? ELP3 protein should be indicated clearly on the MS plot. The axis for proteomics covers -0.3 to 0.3 so a maximum change of 1.23 fold each way. How does the mass spec fold change compare with the western blot fold change for KIF15, KIF4A and KIF14?

Proteomic changes were calculated from the average LFQ intensity of three independent experiments. Log₂ of the delta of log₂ LFQ shELP3 and log₂ LQF control cells (shElp3/shCTR) was then calculated. As the reviewer pointed out, the range resulting from this calculation, is therefore very small. We apologize for this, this was a mistake. We therefore replotted the data by calculating the difference of the average of LFQ intensity (log₂) between shELP3 cells and shCTR (see REV_Table#1, calculation done with Prism graph pad v.8). The new scale is consequently wider and better reflects the changes we observed by western blot. Downregulated and upregulated proteins thresholds were set to -0.5 and +0.5, respectively. As suggested by the reviewer, we highlighted the kinesin members (KIF4A, KIF14, KIF15, KIF23, KIF5B), as well as ELP3 in the revised figure panels presenting the proteomic data (new Fig. 2a and new Suppl Fig. 2a).

Were proteins from Fig1c detected by mass spectrometry?

Among the U₃₄-enriched proteins identified in Fig. 1c, none were detected by MS (REV#2 Fig. 4a). Nevertheless, we could detect the corresponding mRNA expression levels (HMMR, NEXN, MNS1, CEP83 and CEP290) by RNA-seq.

Their corresponding mRNAs, and the mRNA encoding the kinesin members (KIF4A, KIF14, KIF15, KIF23, KIF5B) and ELP3 are indicated in the volcano plot in the new figure (new Suppl Fig. 2b; REV#2 Fig. 4b).

-It is not possible to exclude that changes in protein stability following ELP3 knockdown may explain the absence of changes in KIF5B or KIF23 abundance. Half-life experiments should be performed on at least 1 or 2 U34-enriched candidates.

We performed the experiments suggested by the reviewer. We treated MCF7 cells depleted or not of ELP3 (REV#2 Fig. 4c) with 100µg/ml cycloheximide (CHX) for 4, 8, 16 and 24 hours and we assessed KIF5B and KIF23 protein levels (REV#2 Fig. 4d). Both proteins show similar half-life in control or ELP3-depleted cells. This data excludes that protein stability might explain the absence of changes in KIF5B or KIF23 expression upon depletion of U₃₄-enzymes.

REV#2 Figure 4 legend:

a, b, Volcano plot of the proteomics (n= 3 independent experiments, a) and RNA-seq (n= 3 independent experiments, b) of BT549 cells depleted or not of ELP3. **c, d**, indicated proteins levels were assessed in MCF7 cells depleted of ELP3 (c) upon treatment with 100µg/ml cycloheximide for indicated time points (d).

Reviewer #3 (Remarks to the Author):

In this manuscript, Rapino et al. explained the potential regulatory mechanisms that interconnect translational regulation of the mRNAs and the protein fate/expression levels. To this end, the authors utilized 'loss-of-function models of U34-enzymes' and measured the changes in protein expression level via proteomics and biochemical approaches. To gain a global view on the impact of U34 modification on protein translation, the authors analyzed the total cellular proteome with or without ELP3 knockdown, and monitored changes in protein expression. From the data, authors emphasize that proteins downregulated upon depletion of ELP3 are enriched in U34 codon-bearing genes (while transcript levels remain largely unaltered), although not all U34-codon bearing genes are downregulated (Fig. 1d-e). Importantly, the authors find that kinesin proteins (U34 codon-bearing) are downregulated in protein but not mRNA level, which leads to further dissection of individual kinesin proteins in the following experiments.

Regarding the technical validity of proteomics data, the data interpretation seems to have three major misleading points:

1) *The fold changes in protein expression seem too small (within 0.1 in log₂ scale, according to the Fig. 1d-e and Ext Fig. 2g) to discuss any significant changes in expression level ($< 2^{0.1} = \sim 1.08$; in fact, the RNA-seq expression levels show much bigger changes of $< 2^3 = 8$). This level of fold change should be interpreted as 'No-change', considering the general quantification accuracy of MS intensity based LFQ approach.*

As correctly pointed out by the reviewer, the range of our proteomic is very small. This is a consequence of a mistake in our calculation method. We apologize for the confusion. The proteomic changes were calculated from the average LFQ intensity of three independent experiments. Log₂ of the delta of log₂ LFQ shELP3 and log₂ LQF control cells (shElp3/shCTR) was then calculated resulting in the plotted fold change values.

Agreeing with the reviewer comment, we therefore replotted the data by calculating the difference of the average of LFQ intensity (log₂) between shELP3 cells and shCTR (see REV_Table#1, calculation done with Prism graph pad v.8). The new scale is consequently wider and better reflects the changes we observed by western blot. Downregulated and upregulated proteins thresholds were set to -0.5 and +0.5, respectively. As suggested by the reviewer, we highlighted the kinesin members (KIF4A, KIF14, KIF15, KIF23, KIF5B) and ELP3 in the revised figure panels presenting of the proteomic data (REV#3 Fig. 1a-c; and new Fig. 2a and Suppl Fig. 2a-b). Corresponding mRNAs of the kinesin members, the proteins identified in former figure 1c and ELP3 are now indicated in the volcano plot (REV#3 Fig. 1c; and new Suppl Fig. 2b). All the proteomic related panels have been updated in the new version of the manuscript new Fig. 2a and Suppl Fig. 2a-b).

REV#3 Figure 1 legend:

a, Dot plot of RNA-seq and proteomics of BT549 cells depleted or not of ELP3 (n=3 independent experiments); genes enriched in U₃₄-codons are shown in blue. U₃₄-codons enrichment in downregulated proteins was calculated by χ^2 test. **b, c**, Volcano plot of the proteomics (n= 3 independent experiments, b) and RNA-seq (n= 3 independent experiments, c) of BT549 cells depleted or not of ELP3.

2) The classification of 'Up-regulated', 'No-change', and 'Downregulated' seem a bit skewed (No-change should be of equal width in both directions).

As correctly pointed out by the reviewer, the original graph was misleading. We corrected the plots of the proteomic data clearly setting downregulated and upregulated proteins to -0.5 and +0.5 respectively (new Fig. 2a and Suppl Fig. 2a, REV#3 Fig. 1a-b). We now calculated spearman correlations between codon enrichment (defined as U₃₄-codons cumulative frequency or q-value<0.05) and protein expression (new Fig. 2a; REV#3 Fig. 2a-b, and 2c-d respectively). Our new analyses strengthen all the conclusions of the manuscript.

Codon enrichment calculated as AAA, CAA, GAA cumulative frequency >0.088
(3rd quartile of distribution in the human genome)

Codon enrichment calculated as q-value AAA, CAA, GAA <0.05

REV#3 Figure 2 legend:

a, b, Frequency of U₃₄-codons and protein expression in BT549 cells depleted of ELP3 was plotted, enrichment cut off was defined as 3rd quartile of cumulative frequency distribution Correlation was calculated by Spearman test (a); regression line was plotted (b). **c, d**, Frequency of U₃₄-codons and protein expression in BT549 cells depleted of ELP3 was plotted, enrichment cut off was defined as AAA&GAA&CAA q value<0.05. Correlation was calculated by Spearman test (c); regression line was plotted (d).

3) The western blot results in Fig. 1f shows considerable changes of protein levels for KIF4A/14/15, but would not agree to the proteomics data (if they were identified and quantified; the authors should mark the proteins within the scatter plot).

We highlighted the proteins detected by western blot in figures describing the proteomic data (REV#3 Fig. 1b-c and new Suppl Fig. 2a), the RNA-seq (REV#3 Fig. 1d, new Suppl Fig. 2b) and the combination of the two (REV#3 Fig. 1a, new Fig. 2a).

Regarding the issues raised above, the authors should refrain from making any conclusions regarding protein expression level changes from the “current” proteomics data.

We thank the reviewer for having pointed out the confusion related to the plotting of the proteomics data. We corrected the analysis and replotted the data accordingly. The MS data are in line with the western blot

validation we performed. We now believe that the data, as presented, support the conclusions of the manuscript.

Nonetheless, Fig 1f alone seems sufficient to continue dissection into the roles of U34 codons in translation of kinesin proteins. In this case, however, the authors should take a step back from arguing any global effects of U34 modifications, but put emphasis on the regulation of specific subsets of proteins that have U34 codons (In fact, the author's initial description that some but not all U34-codon bearing proteins are downregulated is itself confusing to discuss the global roles of ELP3).

We thank the reviewer for her/his comments. We now further discussed the initial observation that “that some, but not all U₃₄-codon bearing proteins are downregulated” upon depletion of U₃₄-enzymes. It is commonly accepted that impairment of the mcm⁵s²U₃₄ tRNA modification leads to translation defects due to accumulation of ribosomes on specific codons (i.e. AAA, CAA, GAA) (Nedialkova & Leidel, *Cell* 2015, 161(7):1606-18; Laguesse *et al*, *Dev. Cell* 2015, 35(5):553-567; Rapino *et al*, *Nature* 2018, 558(7711):605-609). Nevertheless, little is known on the impact of the ribosome accumulation on subsequent protein expression levels. It is assumed that the translation defects seen upon impairment of U₃₄-tRNA modification directly translates in changes in corresponding protein outputs. In this work we show that codon-dependent translation elongation defects caused by the impairment of U₃₄-enzymes are not systematically associated with changes in protein expression. While translation elongation defects upon perturbation of U₃₄-tRNA modification are strictly dependent on codon content, the consequences of these defects on protein output are determined by other features. We identify hydrophilic amino acid motifs as essential to dictate protein aggregation and subsequent degradation in contexts of defective translation elongation at U₃₄-codons.

Using the kinesin family as a model to validate this finding, we further demonstrated the causative relationship between the presence of a hydrophilic pentamer and protein aggregation upon depletion of U₃₄-enzymes. Therefore, the combination of the two, a global analysis and a validation using specific examples, allowed us to demonstrate that the presence of hydrophilic pentamers as a key feature to predict U₃₄-enzymes dependent proteome.

We decided to focus on the kinesin family for the following reasons: first, we performed a GSEA analysis, which highlighted that Kinesins were significantly downregulated upon ELP3 depletion in our proteomic dataset (REV#3 Fig. 3a). Second, the kinesin family shows a significant enrichment in proteins whose RNA is strongly enriched in U₃₄-codons compared to human genome (REV#3 Fig. 3b). Third, among the kinesins whose mRNA is enriched of U₃₄-codons, we could highlight difference in behavior upon depletion of ELP3 in our proteomics. As expected, some kinesins enriched in U₃₄-codons, such as KIF4A, KIF14 or KIF15, were downregulated upon ELP3 depletion. Nevertheless, we found kinesins that display similar frequency in U₃₄-codons (i.e. KIF5B or KIF23) but whose expression remained unchanged in the same dataset (Suppl Fig. 2a).

We believe that the conclusions presented in our manuscript using the kinesin members as a model and confirmed using other targets (i.e. NDC80 and NUF2), reflect the discovery of a novel general characteristic essential to understand the role of U₃₄-tRNA modification in protein expression.

REV#3 Figure 3 legend:

a, GSEA analysis of proteomics of the BT549 cells depleted or not of ELP3. **b**, AAA&GAA&CAA cumulative frequency of the members of the kinesin family, red dots depict enriched members (frequency > 0.088). U₃₄-codon rich transcripts enrichment was calculated by χ^2 test compared to the genome.

REVIEWER COMMENTS

Reviewer #1 (Remarks to the Author):

All raised concerns were effectively addressed.

Juana Díez

Reviewer #2 (Remarks to the Author):

In this revised manuscript Rapino and colleagues have made significant changes and improvement to the manuscript. The mass spectrometry data are now presented with a more biologically relevant dynamic range (after the authors realised some data analysis error had been made). The removal of TNBC data make the article a more cohesive story focussed on U34-dependent translation and protein abundance.

The results of the Nanoluc vs FireflyLuc highlight my main concern with this manuscript. It remains unclear when a few U34 dependent codons are sufficient to have a major impact on protein output. The hydrophilic motif may explain a subset of observations, but is not enough to generalize the concept to all transcripts. The reader is then left wondering how many mRNAs enriched in U34-dependent codons do not require the pentasequence to show lower protein abundance following ELP3/CTU2 depletion. This needs to be made abundantly clear by the authors and could be summed up in a table showing how many U34-dependent codon enriched mRNAs have showed lower protein abundance by mass spec without having a pentasequence or with at least one pentasequence. This needs to be done only taking into account the number of proteins identified by mass spectrometry with enough peptide coverage to perform statistical analysis.

Another related point which needs to be clarified is the codon composition of the pentasequence. The three amino acids E, K and R can all be encoded by U34-dependent codons (GAA, AAA and AGA). All three codons are dependent on ELP3, but only GAA and AAA are dependent on CTU1/2. The author should further analyse the "orfeome" to test if the pentasequence is enriched in U34-dependent codon. This becomes extremely relevant in experiments in which xAA codons are recoded in xAG.

Depending on the results obtained from this analysis, experiments should also be performed with KIF4A-mutant construct in which only the pentasequence(s) are changed from XXA to XXG construct to demonstrate to which extent it is the slowing down of translation specifically at the codons encoding the pentasequence or the amino acid sequence on its own (independent of A ending codons) which is responsible for decreased protein abundance/increased aggregation.

The authors mention that DEK, SOX9, HIF1a and hnRNPQ have at least 1 hydrophilic motif? Are the motifs made entirely of "A"-ending U34-dependent codons? The motifs should be provided in supplemental data. Also, figure 4C shows that a large subset of proteins showing no change in aggregation have at least 1 hydrophilic pentasequence. Again, is the codon composition of the pentasequence relevant?

Minor points:

In the abstract, the use of "loss-of-function models of enzymes" suggests that the enzymes have been made catalytically dead while still being expressed, this should be changed to "depletion" or "knockdown".

I still do not agree with the explanation for ranking genes based on all three codons instead of cumulative frequency of each of the XAA codons. Figure 1B clearly shows that each individual XAA codon on its own can affect protein synthesis. There is therefore no requirement to have a certain threshold of all 3 codons to make an mRNA "responsive" to ELP3 or CTU1/2 depletion.

Figure 1D refers to "cumulative frequency of U34 codons" so why not present the same data in 1C for cohesion.

Reviewer #3 (Remarks to the Author):

The authors found a significant mistake in their calculation of mass spectrometric data, which was the origin of the most issues raised by the reviewer, and fixed it in the revised manuscript. Now the authors sufficiently addressed all issues raised. No further concerns remain.

Rapino et al, manuscript NCOMMS-20-20012B

-Point-by-point response to the reviewers' comments-

REVIEWER COMMENTS

Reviewer #1 (Remarks to the Author):

All raised concerns were effectively addressed.

Juana Díez

We thank the reviewer for her positive assessment of our manuscript. We are happy that the revised manuscript addressed all her concerns.

Reviewer #2 (Remarks to the Author):

In this revised manuscript Rapino and colleagues have made significant changes and improvement to the manuscript. The mass spectrometry data are now presented with a more biologically relevant dynamic range (after the authors realized some data analysis error had been made). The removal of TNBC data make the article a more cohesive story focused on U34-dependent translation and protein abundance.

We thank the reviewer for his/her supportive comments.

The results of the Nanoluc vs FireflyLuc highlight my main concern with this manuscript. It remains unclear when a few U34 dependent codons are sufficient to have a major impact on protein output.

The luciferase constructs were designed to include a repetition of U₃₄ codons in frame with the corresponding luciferase cDNA. Different combinations have been tested in our lab. It is important to note that we did not intend to study the mechanisms linking U₃₄-tRNA modifications to protein homeostasis in these experiments. Rather we aimed to test the dependency of codons towards U₃₄-tRNA modification for translation, and to build a cellular read-out tool for the decoding activity of U₃₄-enzymes. We first tried a repetition of 10x XAA codons in frame with the nanoluc. Unfortunately, this setting strongly impacted the expression of the nanoluc, already in control conditions. Therefore, it was discarded. We found that the repetition of 6x successive U₃₄-codons in frame with the nanoluc led to a strong correlation between luc activity and modulation of U₃₄-enzymes expression. Importantly, a repetition of 3 successive codons did not affect the nanoluc expression in a way that it strictly correlated with U₃₄-enzymes expression (also see *Arthur et al, Sci. Adv. 2015 Jul;1(6):e1500154* = PMID 26322332). Again, we did not intend to mimic any physiological regulation in these experiments. Using blasting tools, we could not find any gene presenting a successive repetition of 6x XAA as engineered in these constructs.

As such, in these settings, the depletion of U₃₄-enzymes leads to an extreme ribosome pausing at the 5' end of the ORF, which inhibits the complete synthesis of the luciferase (we did not see aggregation of the reporter construct in these settings). In conclusion, these experiments were designed to confirm the dependency of XAA codons towards U₃₄-tRNA modification for proper translation and to build a cellular read-out for U₃₄-enzymes decoding activity. It should not be

interpreted as a tool to understand physiological regulation of protein synthesis (including the implication of the motifs) by the U₃₄-tRNA modification.

The hydrophilic motif may explain a subset of observations, but is not enough to generalize the concept to all transcripts. The reader is then left wondering how many mRNAs enriched in U34-dependent codons do not require the pentasequence to show lower protein abundance following ELP3/CTU2 depletion. This needs to be made abundantly clear by the authors and could be summed up in a table showing how many U34-dependent codon enriched mRNAs have showed lower protein abundance by mass spec without having a pentasequence or with at least one pentasequence. This needs to be done only taking into account the number of proteins identified by mass spectrometry with enough peptide coverage to perform statistical analysis.

We heard the reviewer comment and we clarified this point in the revised manuscript (new Fig 4d and new SUPPL Table 5). As correctly pointed out, a portion of U₃₄-enriched genes are found downregulated in our proteomics upon depletion of ELP3, without harboring any hydrophilic pentasequence. We made it more clear in the new Fig. 4d. The complete list is now provided in SUPPL TABLE 5. Importantly, the presence of the pentasequence in proteins that exclusively aggregate upon ELP3 depletion is extremely high (new Fig. 4d), supporting our hypothesis that the presence of hydrophilic motifs is a key feature to promote integration of the U₃₄-translational targets into aggregates and subsequent degradation. Therefore, our work highlights the features required to predict protein expression dependency towards the U₃₄-enzymes in the context of aggregation. We do not exclude the existence of other degradation pathways, which likely do not require the presence of the identified motif, to be involved in explaining protein dependency towards the U₃₄-enzymes. We clarified this point in the revised manuscript (lines 41,91-93, 204-205, 242-244 and 335-338).

Another related point which needs to be clarified is the codon composition of the pentasequence. The three amino acids E, K and R can all be encoded by U34-dependent codons (GAA, AAA and AGA). All three codons are dependent on ELP3, but only GAA and AAA are dependent on CTU1/2. The author should further analyse the "orfeome" to test if the pentasequence is enriched in U34-dependent codon. This becomes extremely relevant in experiments in which xAA codons are recoded in xAG.

Depending on the results obtained from this analysis, experiments should also be performed with KIF4A-mutant construct in which only the pentasequence(s) are changed from XAA to XAG construct to demonstrate to which extent it is the slowing down of translation specifically at the codons encoding the pentasequence or the amino acid sequence on its own (independent of A ending codons) which is responsible for decreased protein abundance/increased aggregation.

We agree with the reviewer that this is a very interesting point. We had actually investigated this possibility earlier and we had generated the suggested KIF4A-mutant in which the cDNA sequence of the pentasequence is recoded from XAA to XAG (Fig. REV#2 a-b). Strikingly, we had found that the replacement of the XAA codon into XAG codons within the pentasequence did not affect KIF4A protein levels upon depletion of ELP3, while the complete deletion of the pentasequence or the recoding of the whole cDNA rescued the expression of KIF4A in the same settings (results illustrated in Fig.5c and in Fig. REV#2 a-b).

This excludes the possibility suggested by the reviewer that the codon composition of the identified motif is responsible for decreased protein abundance/increased aggregation upon U₃₄-enzymes depletion.

The authors mention that *DEK*, *SOX9*, *HIF1a* and *hnRNPQ* have at least 1 hydrophilic motif? Are the motifs made entirely of "A"-ending U₃₄-dependent codons? The motifs should be provided in supplemental data. Also, figure 4C shows that a large subset of proteins showing no change in aggregation have at least 1 hydrophilic pentasequence. Again, is the codon composition of the pentasequence relevant?

As required by the reviewer, we provided the amino acid and codon composition of the motifs present in *DEK*, *SOX9*, *HIF1a* and *hnRNPQ* (Fig. REV#2 c). As expected by the significant association between the presence of the motif and enrichment in U₃₄-codons (Supplementary Table 3, p=9,84E⁻⁴⁸), K, E and R in the pentamers are also encoded by U₃₄-codons. Specifically, the U₃₄-codon usage in the pentamers is: AAA= 51%, GAA=57% and AGA=11% compared to genome codon usage of 43%, 42% and 21%, respectively. Nevertheless, our experiment replacing the XAA codon into XAG codons within the pentasequence of KIF4A discharges the hypothesis that the pentasequence codon composition plays a role in determining protein fate upon U₃₄-enzymes depletion. Rather, we demonstrated that the ampholytic nature of the hydrophilic pentasequence is responsible for protein aggregation upon U₃₄-enzymes dependent translation elongation defects.

The reviewer correctly highlights the presence of pentasequences in proteins whose expression does not change upon ELP3 depletion (Fig. 4c). As before, we believe that codon composition of the pentasequence is not relevant to understand why these proteins are not affected by the depletion of the U₃₄-enzymes. Several additional mechanisms could explain this observation (i.e: binding of other proteins, localization of the motif in specific domains, ...). Further studies would be needed to clarify this aspect.

Figure Rev.2. **a**, Scheme of the KIF4A mutants, motif is depicted in red. **b**, MCF7 cells overexpressing the indicated KIF4A mutants and depleted of ELP3. Protein levels were assessed by western blot. **c**, Table showing the hydrophilic pentasequences of the listed proteins and their corresponding mRNA sequence. U₃₄-codons are depicted in bold.

Minor points:

In the abstract, the use of “loss-of-function models of enzymes” suggests that the enzymes have been made catalytically dead while still being expressed, this should be changed to “depletion” or “knockdown”.

This has been changed accordingly (line 33)

I still do not agree with the explanation for ranking genes based on all three codons instead of cumulative frequency of each of the XAA codons. Figure 1B clearly shows that each individual XAA codon on its own can affect protein synthesis. There is therefore no requirement to have a certain threshold of all 3 codons to make an mRNA “responsive” to ELP3 or CTU1/2 depletion.

Figure 1D refers to “cumulative frequency of U34 codons” so why not present the same data in 1C for cohesion.

As stated earlier, the results shown in panel 1B only show the dependency of codons towards U₃₄-tRNA modification enzymes for translation. This setting is not meant to reflect physiological relevance as we used artificial successive repetition of specific codons, a situation that is never seen in genes. We believe that this is an overinterpretation of the data to claim that only individual XAA codons can affect protein synthesis.

The method we developed here to identify genes with different codon composition is indeed very stringent and based upon the calculation of q-values. We wish to stick to this strategy that was used to identify genes with an extreme dependence towards U₃₄-tRNA modification and enzymes. This was confirmed by the results showed in figure 1d-e in which we found that this strategy was very relevant to highlight strong dependency of identified genes towards U₃₄-enzymes. We now provide the list of genes with codon enrichment and the presence of hydrophilic motifs as requested in the previous comment of the reviewer (new Supplementary Table 3). We believe that overall, the data provided are clear and accessible to the reader, depending on the question one would like to address.

Reviewer #3 (Remarks to the Author):

The authors found a significant mistake in their calculation of mass spectrometric data, which was the origin of the most issues raised by the reviewer, and fixed it in the revised manuscript. Now the authors sufficiently addressed all issues raised. No further concerns remain.

We thank the reviewer for his/her positive assessment of our manuscript. We are happy that the revised manuscript addressed all his/her concerns.

REVIEWERS' COMMENTS

Reviewer #2 (Remarks to the Author):

With this revised manuscript, the authors have now addressed all my remaining concerns.